# A multicentre study on spontaneous in-cage activity and micro-environmental conditions of IVC housed C57BL/6J mice during consecutive cycles of bi-weekly cage-change

B. Ulfhake[1]*, H. Lerat[2], J. Honetschlager[3], K. Pernold[1], M. Rynekrová[3], K. Escot[2], C. Recordati[4,5], R. V. Kuiper[6,7], G. Rosati[8], M. Rigamonti[8], S. Zordan[8], J.-B. Prins[9,10]

1 Department of Laboratory Medicine, Karolinska Institutet, Stockholm, Sweden, 2 Université Grenoble-Alpes, UMS hTAG Inserm US046 CNRS UAR2019, Grenoble, France, 3 Institute of Molecular Genetics of the Czech Academy of Sciences, Prague, Czech Republic, 4 Department of Veterinary Medicine and Animal Sciences, University of Milan, Lodi, Italy, 5 Mouse and Animal Pathology Laboratory, Fondazione Unimi, Milano, Italy, 6 Department of Laboratory Medicine, Karolinska Institutet, Huddinge, Sweden, 7 Norwegian Veterinary Institute, Section Aquatic Biosecurity Research, Oslo, Norway, 8 Tecniplast SpA, Buguggiate (Va), Italy, 9 Central Animal Facility, PDC, Leiden University Medical Centre, Leiden, The Netherlands, 10 The Francis Crick Institute, London, United Kingdom

* brun.ulfhake@ki.se

**Data Availability Statement:** The data relevant to this study are available from the Dryad repository at DOI: 10.5061/dryad.ksn02v75q.

## Abstract

Mice respond to a cage change (CC) with altered activity, disrupted sleep and increased anxiety. A bi-weekly cage change is, therefore, preferred over a shorter CC interval and is currently the prevailing routine for Individually ventilated cages (IVCs). However, the build-up of ammonia ($NH_3$) during this period is a potential threat to the animal health and the literature holds conflicting reports leaving this issue unresolved. We have therefor examined longitudinally in-cage activity, animal health and the build-up of ammonia across the cage floor with female and male C57BL/6 mice housed four per IVC changed every other week. We used a multicentre design with a standardised husbandry enabling us to tease-out features that replicated across sites from those that were site-specific. CC induce a marked increase in activity, especially during daytime (~50%) when the animals rest. A reduction in density from four to two mice did not alter this response. This burst was followed by a gradual decrease till the next cage change. Female but not male mice preferred to have the latrine in the front of the cage. Male mice allocate more of the activity to the latrine free part of the cage floor already the day after a CC. A behaviour that progressed through the CC cycle but was not impacted by the type of bedding used. Reducing housing density to two mice abolished this behaviour. Female mice used the entire cage floor the first week while during the second week activity in the latrine area decreased. Measurement of $NH_3$ ppm across the cage floor revealed x3 higher values for the latrine area compared with the opposite area. $NH_3$ ppm increases from 0–1 ppm to reach ≤25 ppm in the latrine free area and 50–100 ppm in the latrine area at the end of a cycle. As expected in-cage bacterial load covaried with in-cage $NH_3$ ppm. Histopathological analysis revealed no changes to the upper airways covarying with recorded $NH_3$ ppm or bacterial load. We conclude that housing

**Funding:** The work at IMG was funded by IMG. The work at UGA was funded by UGA. The work at KI was funded by Karolinska Institutet and the Swedish National Research Council (Grant 2020-02009-3). The work at LUMC was funded by the LUMC. DVC® equipment at LUMC and UGA was made available by Tecniplast SpA. S. Zordan, M. Rigamonti and G. Rosati were employed by Tecniplast SpA, which provided support in the form of salaries. Tecniplast SpA and the other funding bodies did not have any additional role in the study design, data collection and analysis, decision to publish, or preparation of the manuscript. The specific roles of the authors are articulated in the 'author contributions' section.

**Competing interests:** The authors declare no conflict of interest. Tecniplast SpA (Via I Maggio 6, 21020 Buguggiate (Va), Italy) is a commercial company selling the DVC® system. However, this does not alter the authors' adherence to all the PLOS ONE policies on sharing data and materials. We have read the journal's policy and the authors of this manuscript have no competing interests. The data recorded at each site remains the intellectual property of respective site.

of four (or equivalent biomass) C57BL/6J mice for 10 weeks under the described conditions does not cause any overt discomfort to the animals.

## Introduction

The mouse (*Mus musculus*) is the species most used in biomedical research. The cage environment and husbandry procedures are designed to ensure welfare and health of the animals, which in their turn impact experimental outcome and reproducibility and may introduce confounding effects [1–6]. In the care of small rodents, the recurring cage change is a routine procedure to prevent over soiling of the internal cage milieu. The interval between subsequent cage changes is the subject of a large body of studies considering different types of housing systems [7–24], housing density [7, 13, 19, 20, 24–26], strain, age and sex of the animals [13, 14, 19, 20, 24, 25, 27], and bedding materials used [16, 17, 23, 25, 28–31]. In several studies, the build-up of ammonia ($NH_3$) during the period between cage changes has been reported as a critical parameter [9, 10, 13, 14, 16, 17, 19–21, 23–25, 27, 28, 31–34], because of its potential threat to animal health [7, 9, 13, 16, 20, 21, 24, 25, 35–38] (see also Toxicological Review of Ammonia Noncancer Inhalation [CASRN 7664-41-7] Supplemental Information, September 2016).

However, the issue of the extent of that threat remains unresolved. The outcomes reported from facility studies with similar type of holding systems (e.g. different types of individually ventilated caging systems, IVCs) are not consistent except for static microisolators where the reports are in general agreement [16, 17, 21, 24, 33]. Some studies report low $NH_3$ values for IVC systems even with a prolonged cage-change interval of 2, 3 or even 4 weeks, while others observed higher $NH_3$ values even with weekly changes [7, 9, 11, 13–17, 19, 20, 22, 23, 39]. A separate issue is the question of the safe exposure limit for small rodents [11, 13, 21, 22], and if the commonly adopted human work-place exposure limit is relevant for mice (25 ppm on average across 8h with accepted peak values of 50ppm; see EPAs Toxicological Review of Ammonia Noncancer Inhalation [CASRN 7664-41-7] Supplemental Information, September 2016)? Studies looking into in-cage microenvironment and reporting histopathological analysis of the animals are inconsistent and across studies histopathological observations do not seem to correlate with observed in-cage $NH_3$ ppm (c.f. [7, 16, 20, 24, 33, 40]). As reviewed elsewhere, the different techniques used for assessing in-cage $NH_3$ ppm may explain some of these differences [41]. Questions remain to what extent the recorded $NH_3$ ppm:s correspond to the real exposure of the animals and if the histological stigmata observed can be ascribed to environmental $NH_3$ exposure with certainty [33]. In the facility studies cited above, the $NH_3$ ppm was most often obtained with a probe inserted through a passage in the cage wall taking a point measurement at a level 1–3 cm above the bedding (*idem*). The sampling was then repeated on a daily, weekly or once per cage change basis and used as a proxy for the real exposure of the animals (*idem*). A single point in-cage $NH_3$ measurement may not be sufficient considering the variance in $NH_3$ ppm across the cage floor [16, 42] and combined with the different $NH_3$ detection technique used, help to explain the lack of consistency between studies as well as the disagreement with histopathological observations of the airways in small rodents after controlled acute and chronic exposure to $NH_3$ (nose-only or chamber with clamped $NH_3$ ppm [33, 38, 40]).

Although most facility studies on in-cage microenvironment including those with extended cage-change period report different health outcomes, data on the spontaneous in-cage pattern

of rest and physical activity is missing. We recently published cumulative records of the spontaneous in-cage activity of female C57BL/6J subjected to a weekly cage-change, including the impact of cage-change and other husbandry routines [2]. Although the cage change results in a decrease of the in-cage load of mould and bacteria, hence $NH_3$ concentration (ppm), and an increase of soil-free cage floor space [43, 44], the animals react with altered behaviours and responses, increased hart rate and mean arterial blood pressure, disrupted sleep patterns, and increased levels of stress hormones [1, 2, 9, 45–51].

We report here on a multicentre study where we recorded and analysed cumulative records of spontaneous in-cage activity and $NH_3$ ppm across the cage floor during five consecutive bi-weekly cage-change cycles. The four participating centres used identical husbandry protocols, and both male and female C57BL/6J mice were kept in digital ventilated cages, i.e. IVCs complemented with an external sensor-board enabling collection of in-cage acitivity data 24/7 (DVC®; Tecniplast S.p.A.) [2, 52]. Three times a week, ammonia concentrations were measured under flow conditions across the front, middle, and rear of each cage. At one centre, additional experiments were performed assessing in-cage bacterial load, varying cage-change intervals, bedding materials, and numbers of animals per cage. At the end of the experiment, tissues of the upper airways from one randomly chosen animal from each cage was analysed for histopathological changes (n = 20). As a reference served germ-free (GF) animals of the same strain, sex and age, as GF mice are known to have lower endogeneous levels of $NH_3$ and are devoid of bacteria responsible for generating $NH_3$ from urea in the cage environment [53–55].

## Materials and methods

### International centres

This study was conducted at four European centres during summer-early autumn 2018: Karolinska Institute, Sweden (KI site), Leiden University Medical Center, The Netherlands (LUMC site), Institute of Molecular Genetics of the Czech Academy of Sciences, Czech Republic (IMG site), Université de Grenoble Alpes, France (UGA site). Three centres (KI site, LUMC site and UGA site) recorded animal locomotion data and performed ammonia measurements while IMG site recorded in-cage activity only.

### Ethical considerations

The care and use of all mice in this study were under national animal protection regulations in accordance with EU Directive 2010/63/EU. They were approved by the respective institutional Animal Welfare Committees (AWB).

License no. KI: by the Regional Ethics Council (Stockholms Djurförsöksetiska nämnd) project license N116-15

License no. LUMC: the study plan 'Ammonia detection in IVC cages in a DVC® rack with mice under flow conditions during normal husbandry and care procedures' was approved by the LUMC Animal Welfare Body

License no. IMG: 66866/2015-MZE-17214 from the Ministry of Agriculture

License no. UGA: C 38 516 10 006, from Direction Départementale de la Protection des Populations.

### Animals and husbandry

In this study, all centres used specific pathogen-free (SPF) C57BL/6J mice (*Mus musculus*) of both sexes, delivered by commercial breeders at the age of 6–8 weeks (Table 1). The SPF status

**Table 1. Husbandry routines.**

| | *IMG* | *KI* | *LUMC* | *UGA* |
|---|---|---|---|---|
| **Start of the Project** | 14th August 2018 | 20th June 2018 | 20th July 2018 | 11th September 2018 |
| **Animal strain** | C57BL/6JOlaHsd (Envigo, Italy) | C57BL/6J (Charles River, Germany) | C57BL/6J (Charles River, Germany) | C57BL/6JRj (Janvier Labs, France) |
| **Animal identification** | Ear tag (National Band & Tag Co) | Ear punch | Ear punch | Ear punch |
| **Nesting** | Bed-r'Nest (Datesand) | Bed-r'Nest (Datesand) | Bed-r'Nest (Datesand) | Bed-r'Nest (Datesand) |
| **Bedding** | Aspen 5x5x1mm– 100g (Tapvei) | Aspen 5x5x1mm– 100g (Tapvei) | Aspen 5x5x1mm– 100g (Tapvei) | Aspen 5x5x1mm– 100g (Tapvei) |
| **Cage Change method** | Tail | Forceps prevailing or hand | Tail | Forceps |
| **PPE** | Gloves, face covering, hair cover, gown | Gloves, no face covering, hair cover, gown | Gloves, face covering, hair cover, gown | Gloves, face covering, hair cover, gown |
| **Rotation of cages after cage change** | Yes | Yes | Yes | Yes |
| **Diet** | SDS R3 (irradiated) | SDS R3 (irradiated) | SDS R3 (irradiated) | SDS R3 (irradiated) |
| **Water** | RO chlorinated water in bottles. Water bottle changed every week | Weakly chlorinated tap water. Water bottle changed every week | Autoclaved, non-chlorinated tap water. Water bottle changed every week | RO[1] water filtered (0.3μ). Water bottle changed every week |
| **Dark-Light Cycle hours** | Dark period 4:00 pm—4:00 am with 10min of dawn and dusk | Dark period 4:00 pm—4:00 am without dawn and dusk | Dark period 7:00 pm—7:00 am with 30min of dawn and dusk | Dark period 7:30 pm—7:30 am without dawn and dusk |
| **Light Levels [lux]** | A rack column: | A rack column: | A rack column: | Only F rack column: |
| **mean (1st-3rd quartiles)(min-max)** | 66(58–68)(57–78) | 32 (21–40) (17–57)[2] | 11(8–11)(7–21) [2] | 20(14–28)(12–34)[2] |
| | F rack column 45(42–46)(39–54)[2] | F rack column: 33 (23–42) (16–59)[2] | F rack column: 11(8–11)(7–21)[2] | |
| **Room T and R.H. [˚C] [%]** | T within 22–24 ˚C | T = 20.2 (20.1–20.4)(19.3–22.6)[2] | T = 21.0 (21.0–21.1)(19.7–22.5)[2] | T = 21.7 (21.2–22.2)(19.5–23.5)[2] |
| **Mean(1st-3rd quartiles)(min-max)** | R.H. within 40–60% | R.H. = 59 (52–63)(44–75)[2] | R.H. = 47 (42–56)(38–64)[2] | R.H. = 40 (31–50)(17–64)[2] |

[1] Reverse osmosis.

[2] Order of values: mean (1st—3rd quartile)(minimum—maximum values).

at all four institutes was defined as to exclude all the recommended infectious agents to monitor for laboratory mice listed by the FELASA recommendations for health monitoring [5, 6].

Mice were observed daily by the animal care staff for signs of ill-health and compromised wellbeing according to local requirements by national regulations under the EU Directive 2010/63/EU [56]. Animals were weighed every other week at the time of the cage-change.

According to the centre's standard routine, the animals were marked by ear punch or ear tag upon arrival. Separated by sex, the animals were randomised (for details see S1 File, Randomisation) to different groups and housed 4 mice per cage on a single rack (DVC® rack, Tecniplast). The mice were kept in modified (see in the DVC® equipment section for further details) Individually Ventilated Cages (IVC GM500, Tecniplast SpA), equipped with DVC® boards positioned underneath the cage and fixed to the rack.

All cages were prepared with 100g of autoclaved aspen chip (AC) bedding (Tapvei, 5x5x1mm) and provided with one puck of Bed-r'Nest as nesting material (Datesand). The bedding quantity was based on the target volume consistent with Tecniplast's standard requirement practice. These requirements are defined as to reach the best compromise between air ventilation efficiency based on GM500 air-flow dynamics, sufficient bedding depth to provide

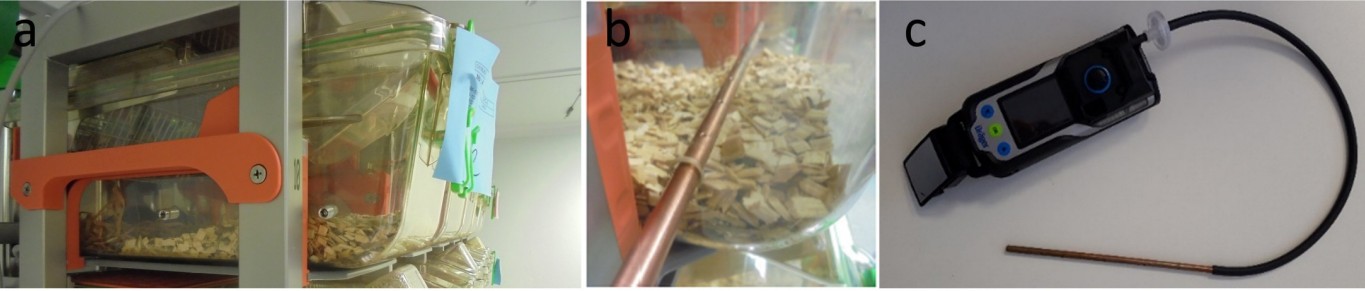

**Fig 1.** a) Modified IVC GM500 cage with holes drilled in the long side, sealed with screws and nuts inserted into the DVC®rack. b) Lateral view of the modified IVC GM500 cage with the NH3 detector probe inserted across the width of the cage to perform the ammonia measurement. c) Dräger X-am® 8000 device with a 40 cm long rubber tube, dust filter and copper probe.

the animals with the opportunity to perform digging activities, and to reduce bacterial growth as much as possible under animal holding room conditions as per Annex 3 of 2010/63/EU. Likewise, based on the manufacture's advice the amount of corn cob (CC) used as alternative bedding at KI (Bed-o'Cobs ¼" from Datesand) was set at 200gms per cage.

All animals were fed *ad libitum* with irradiated diet (Standard Diet RM3, Special Diet Service), and had *ad libitum* access to Reverse Osmosis (RO) chlorinated water, weakly chlorinated tap water or autoclaved tap water depending on the centre. At all centres, the holding room temperature (T) and relative humidity (RH) were set to keep relative humidity and temperature within the legally defined ranges as per Annex 3 of 2010/63/EU (Table 1). Mice were kept in a 12h light/dark cycle, with or without dawn and dusk dependent on the holding room standard of the centre (Table 1). The cage change procedure was performed every second week and carried out by forceps or tail handling depending on the centre's routine. All the relevant husbandry information is reported in Table 1. Recorded time of day of the five cage changes at the different sites is shown in Fig 1 of S1 File.

## DVC® equipment

All centres were equipped with the same type of DVC® rack (Tecniplast SpA) with 60 cage-positions. The equipment is a standard IVC rack with electronic boards in all cage positions. It enables automated, nonintrusive 24/7 recording of spontaneous in-cage floor (500 $cm^2$) activity [2, 52].

All cages were individually ventilated by an Air Handling Unit (SmartFlow blowers, Tecniplast SpA), providing HEPA-14 filtered air to all inserted cage at 75 air changes per hour (ACH) and negative pressure inside the cage (-20%, negative mode).

All centres were provided with modified IVC cages with three holes drilled 1.5 cm above the bedding level along the cage's lateral wall corresponding to the rear, middle, and front area of the cage floor. When not in use, each hole was sealed by a screw and a nut (Fig 1a).

## Ammonia (NH₃) concentration (ppm) measurements

The recording of ammonia levels, i.e. $NH_3$ ppm values, from every cage while inserted in the rack under airflow conditions, was carried out using an industrial ammonia detector device (Dräger X-am® 8000, Dräger) with an electrochemical sensor connected via a dust filter and a 40 cm long rubber tube to the open end of a copper probe with multiple perforations at equal distance along its axial dimension. The copper probe's length was designed to extend across the full length of the cage's shorter planar dimension when introduced from the side of the

modified cage (see Fig 1b and 1c). According to the supplier datasheet, the Dräger X-am® 8000 has a dynamic range spanning from 0 to 300 ppm of $NH_3$. All detectors were calibrated by Dräger immediately before the start of the experiment. Methods and technology used to measure intracage $NH_3$ levels have been reviewed by Morrow et al. [41]

## Experimental protocol

On the day of arrival, animals were randomly assigned to IVC cages to form 10 groups of 4 male mice each and 10 groups of 4 female mice each(for details see above). The 10 cages with male animals and the 10 cages with female animals were inserted in alternating rows of the lateral (A and F) columns of the DVC® rack (Fig 2). This lateral placement was needed to allow insertion of the ammonia detector probe from the cage's lateral side to perform each $NH_3$ measurement under flow condition. The central (B-E) columns were filled with cages without animals, but with bedding and enrichment similar to the cages with animals. The animals were allowed to acclimatise to their new environment for 14 days.

Full cage changes (cage body plus cage top) were performed every 14 days for five consecutive cycles resulting in a total of ten weeks. Neither dirty bedding nor part of the used enrichment was transferred from the dirty to the clean cage. Between two cage changes, operators took ammonia measurements inside the cages at six instances for LUMC site (days 2, 4, 6, 8, 11, 13 just before the next cage change procedure) and five for the others (days 2, 5, 7, 9, 13). For a detailed protocol of $NH_3$ measurements see S1 File.

During each cage change procedure, animals were weighed as one aspect of animal welfare surveillance.

To mitigate possible effects due to different light levels along the vertical axis of the rack columns used, each cage was moved two positions down at each cage-change. The bottom two cages on each side of the rack were moved to the top two positions (see Fig 2b).

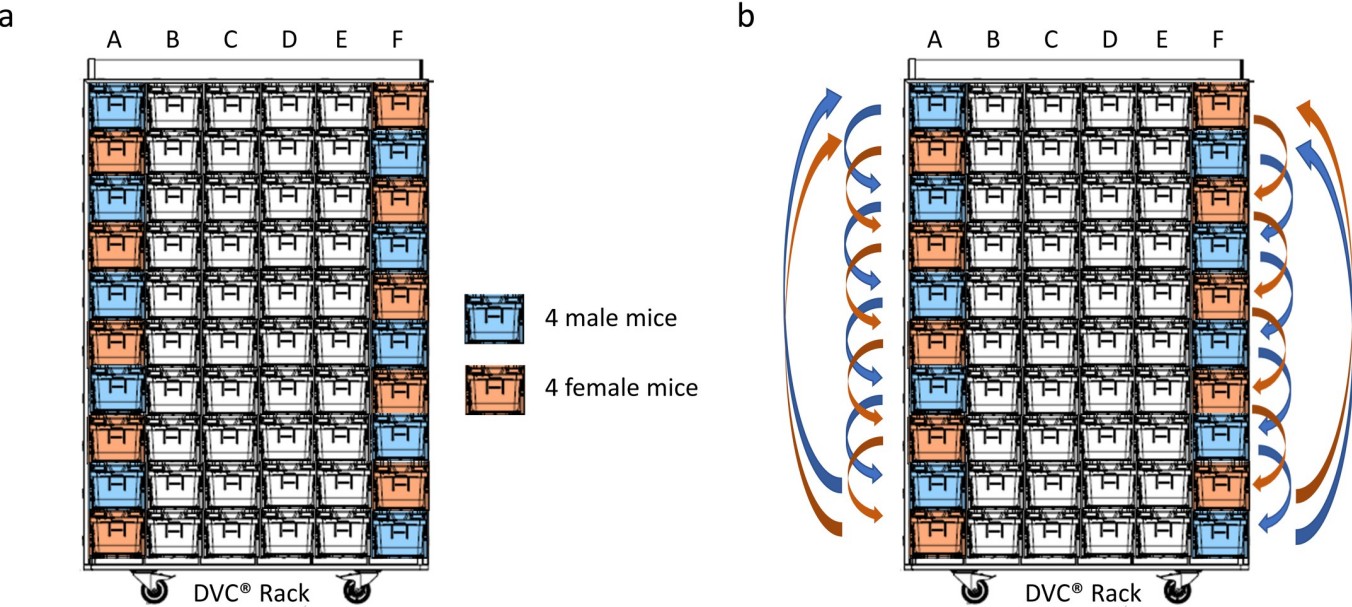

**Fig 2.** a) Lateral DVC® rack columns were alternately filled with cages with female (orange) and the cages with male mice (light blue). The central (B-E) columns were filled with empty cages (white). b) Cage movement scheme at each cage change.

## In-cage bacterial growth in-between cage-changes and tissue harvesting for post hoc histopathological examination

At the KI site, all cages were swabbed for bacterial load assessment. Swabbing was done at the end of the five bi-weekly cage-change cycles, at the end of the two cycles of weekly cage change and at the end of the second cycle, each cage was swabbed once more. Swabbing included all four walls and corners at the level of the bedding material. After swabbing, each swabs was stored in media from the service provider Eurofins (Swedish branch). According to instructions from the service provider, the swabs in media were shipped to Eurofins for analysis of aerobic microorganisms at 30°C (LPP0C-1, UM4PK-1) according to AFNORM (Certificate no 3M 01/01-09/89; by Eurofin Food and feed Testing, Sweden) within 24 h at ambient temperature. Results are in colony-forming units (CFU) and analysed after log conversion to normalise the data.

Twenty animals (ten males and ten females), one from each cage, were randomly selected, and euthanised, followed by dissection of the upper airways for histopathological analysis. Briefly, the animals were decapitated under isoflurane anaesthesia. Skull and brain together with lower jaw were removed, and the remaining specimen was irrigated with 4% buffered (pH 7) paraformaldehyde (PFA) [24, 57]. The tissue was kept in 4% PFA for 24 hours at 4°C and then transferred to 70% ethanol and stored at 4°C until decalcification in 85% formic acid/citrate (pH 2.2) until soft. Coronal sections of the nasal cavities at different levels were dissected and embedded in paraffin. Paraffin blocks were cut in 4 μm sections on a rotary microtome. Sections were mounted on coded slides, and stained with haematoxylin and eosin (HE) for analysis by light microscopy. Histopathological analysis was performed in compliance with previously published protocols [13, 24, 57] and executed blinded and independently by two histopathologists (CR and RK).

As a reference of mice with a facultative low exposure to $NH_3$ [58, 59] served twenty, age and sex-matched C57BL/6J mice raised, maintained and shipped as germfree (GF) (Animal Research Center, Ulm University, Oberberghof, Ulm, Germany). See (S1 File, Germ free mice) for details on the generation and maintenance of the GF colony at the Ulm center. GF mice have been shown to produce less $NH_3$ in their intestines and to have lower serum levels than mice with microbiota [53]. Importantly, urea excreted with the urine deposites will not be catalyzed to $NH_3$ in a germ-free environment [54, 55]. Upon arrival in Stockholm, Sweden, the animals were immediately euthanised by decapitation while under isoflurane anaesthesia, dissected and the tissue processed as described above. Upon breaking the cylinder seals, materials from each transport cylinder was secured by aseptical technique and sent back to the service provider for analysis to confirm the germ free status upon arrival at KI.

Seven different histopathological entities (Table 2) were ranked on a scale from not present (0) to severe (5). From each animal, one serial section (n = 40) was immunostained with cleaved-Casapase 3 (asp175) (cat. No. 9661, Cell Signaling Technologies) for the assessment of apoptosis [60]. The number of cleaved-Casapase-3 positive cells were counted in the olfactory and respiratoy epithelium in the half upper portion of the nasal cavities, and their number was normalised for the length of the evaluated epithelium. In total, nine variables were included in the analysis (Table 2).

## Impact of bedding material and housing density on in-cage activity and $NH_3$ levels

At the KI site, additional experiments using different bedding material and different housing density were carried out during late autumn and winter 2018/2019. Specifically, two groups of ten cages with four male mice in each (same strain, breeder, and age as in the main study

**Table 2. Histopathological variables and anatomical location.**

| Location | Stigmata | Reporting | Interpretation |
|---|---|---|---|
| **Nasal septum** | Accumulation of "amyloid" | Background, common spontaneous nasal lesion, age-related [13] | Chronic; not reversible? |
| **Respiratory epithelium (RE)(Septum, turbinates)** | Hyalinosis | | Chronic; reversible |
| **Olfactory epithelium (OE)(dorsal meatus)** | Hyalinosis | Spontaneous change, age-related [13] | Chronic; reversible |
| | Apoptotic cells | Possibly pathological | Acute; not reversible |
| | Inflammatory cell infiltrate | [13, 24] | Acute/subacute/chronic; reversible |
| **Other** | Nasal gland cysts | [13] | Acute/subacute; reversible |

described above) were housed on either aspen chip (AC) bedding or corn cob (CC) bedding (200g of Bed-o'Cobs ¼" from Datesand). Cages with AC and CC, respectively, were inserted in every other slot of the lateral columns of the DVC® rack. These 20 cages underwent the same procedures explained in the previous section, but this time only for three consecutive cycles of bi-weekly cage-change. During the cage change at the end of the third cycle, the housing density was lowered to two animals per cage. Another two complete cycles were performed to collect animal activity and $NH_3$ measurements during the bi-weekly regime.

## DVC® data processing, data computations, and statistical analyses

DVC® metrics. We used activation density metric to measure animal activity in the home cage [52]. Data (raw data resolution is one sample per 250 msec) were aggregated in bins of one minute. We considered day time activity (average of all one minute bins within the lights-on period of each day), night time activity (average within the lights-off period) and, as previously reported [2], diurnality, which is the percentage of daily activity performed during lights-on over the total activity during the 24 hours day (considered as the sum of lights on and lights off activity). Nocturnal species like *Mus musculus* typically display a reversed circadian rhythm [2]. We then analysed activity in the rear (by averaging activity measured by electrodes 1-2-3-4-5-6) and in the front (electrodes 7-8-9-10-11-12) areas (250 cm$^2$ each) of the cage separately (Fig 3). We decided to divide the cage floor into two parts and not in multiple minor areas, based on the records of latrine position (see below and the Results section). This allowed us to calculate Frontality, which is the percentage of activity performed in the front part of the cage over the total activity performed across the cage's entire floor. We used this metric to assess the spatial distribution of activity over time in relation to $NH_3$ measurements and latrine position.

Since raw DVC® capacitance readings are affected by the presence of water and therefore urine [52], we used the DVC® technology to identify the position of the latrine inside the cage. We determined the latrine position as the area with the highest difference between the average of capacitance readings of the last night of the cycle and the average over the first night of the cycle (for further information see Fig 5 in S1 File). This procedure was validated towards measured ppm $NH_3$ and the standard routine where latrine position is determined by visual inspection at cage-change.

It should be noted that since the animals were group housed, all activity data presented in this study are the collective floor activity of all mice in the cage.

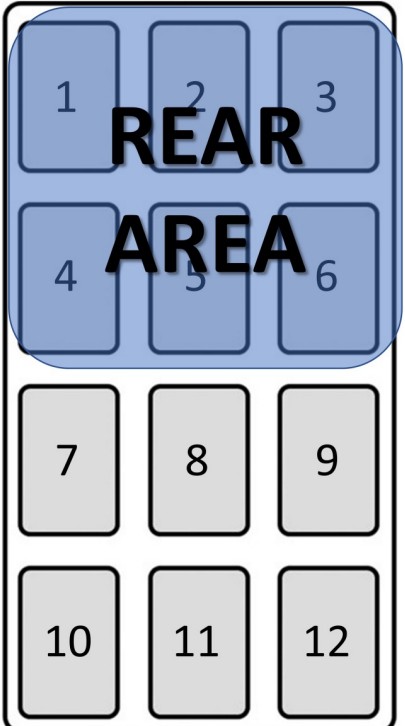 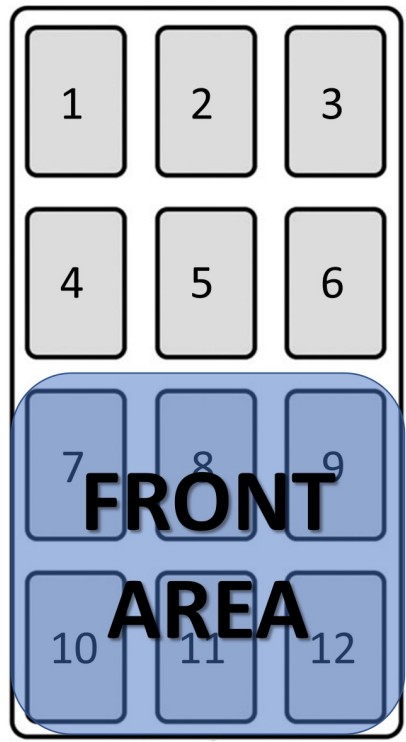

**Fig 3. DVC® board with 12 electrodes.** The rear half of the cage (Rear Area) is covered by electrodes 1-2-3-4-5-6. The front half of the cage (Front Area) is covered by electrodes 7-8-9-10-11-12.

## Statistical analysis

To test differences across sites or sexes, cage-change cycles, and days, we performed repeated measures analysis, using the rank-based analysis of variance-type statistic (ATS), as implemented in the nparLD R Software package [61, 62]. We chose a non-parametric test instead of Repeated Measures ANOVA because normality assumptions were violated in some cases. We considered cages as subjects; time, event, and observation-order as within-subject factors ("sub-plot" repeated factors), and sex and site as between-subject factors ("whole-plot" factor). According to authors' terminology [55], we used either F2-LD-F1 (sex and site as whole plot factors and observation order as the repeated factor) or F1-LD-F2 (sex as the whole-plot factor and event and observation order as the repeated ones), and F1-LD-F1 or LD-F1 when only observation order was used as subplot factor. The statistical analysis of time-series with nparLD is based on rank-order of the observed data, with the relative effect size ($p_s$) as effect size measure [62]. The difference towards parametric tests being that instead of the mean difference between observations, the rank-order is used to assess the probability that two sets of observations differ ($p_s = 0.5$ means that there is no difference in rank-order), and with longitudinal data if the relative effect size varies across the time-series within the sets of observations.

Comparison of two independent samples, and paired samples, were conducted using non-parametric Mann-Withney U statistics (U-test and Wilcoxon's test for matched pairs). In these instances we used the common language (CL) effect size statistics [63–66]. The CL effect size is based on the rank-order (rank sum) of the observed values, and indicates the relative frequency with which the rank sum from one set of observations will be larger than the rank sum of a second set of observations.

Correlation was conducted with the nonparametric Spearman rank correlation. The Spearman correlation coefficient ($r_s$) indicates the effect size with a range from a perfect inverse covariation ($r_s = -1$), through no covariation ($r_s = 0$) to a perfect positive covariation ($r_s = 1$) of the ranks for two parameters.

We used Python to process and visualize data and R to run the statistical tests.

Histopathological analysis by two independent specialists of coded slides from 20 animals (10 of each sex) and 20 germfree animals (same age, both sexes) of the nostril region, middle and rear part of the upper airways included ranking the prevalence (from 0 = negative to 5 = prevailing) of a range of parameters in each of the three regions (see Table 2). This set of ordinal data from each of the three regions was first analysed using multiple covariance analysis (MCA) [67] controlling for sex and axenic status, and subsequently, run against the in-cage $NH_3$ ppm and bacterial load records (continuous data) using a mixed model principle component analysis (mixPCA) [68, 69].

## Results

### Animal health and body weight

During the study, no animal was lost at any of the sites due to unexpected death or removal because of ill-health. Bodyweight time series (Fig 4) showed growth rates of both males and females comparable between sites and to those published for C57BL/6J.

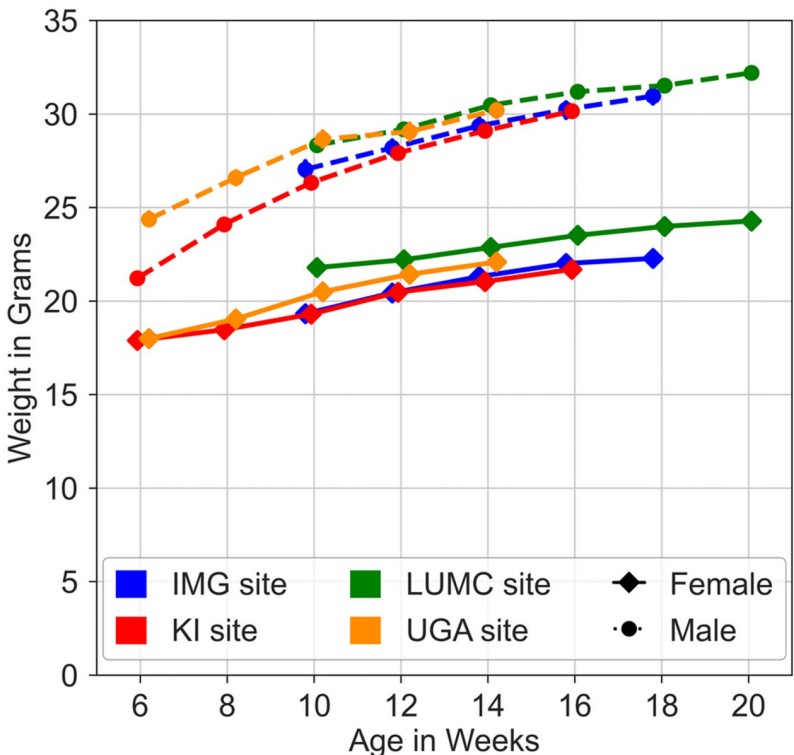

**Fig 4. Body weight time series for male and female C57BL/6J mice at the four sites.** The different sites have been indicated by colour (see the key to symbols), male data are represented by stippled lines while female data are represented by continuous lines. Data (dots) are mean values ± SEM for each sex and site.

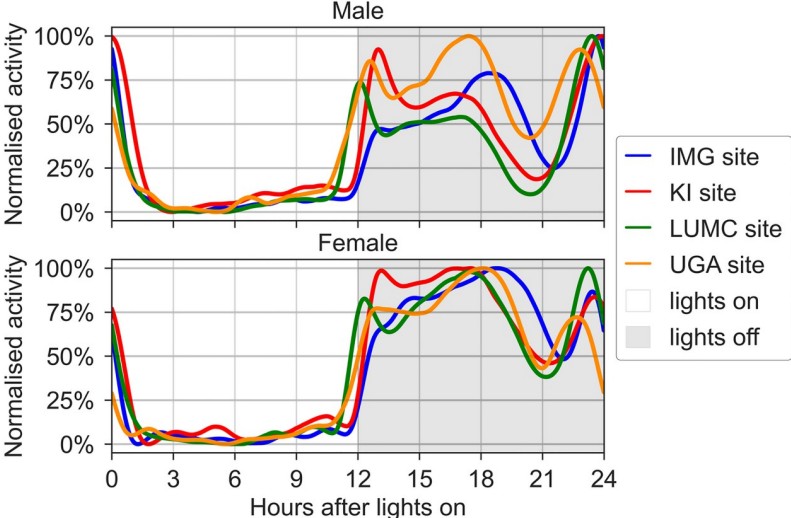

**Fig 5. Circadian rhythmicity of in-cage activity of male (upper panel) and female (lower panel) mice at the four sites.** The panels show the relative distribution of activity for male (upper panel) and female (lower panel) mice during lights on (ZT 0–12, day) and light off (ZT 12–24, night) (ordinate is % of maximum activity at each site across all cages and cage-change cycles). In greater detail, the graphs represent the average activity pattern across the 24 hours by considering the activity time series (1,440 minutes) for each cage and day (except the day of cage-change), smoothing it with a moving average of 60 minutes and normalising to peak activity [2].

## Circadian rhythmicity of activity during lights-off and lights-on

Most laboratory mice strains like C57BL/6J are nocturnally active and rest during the lights-on phase [70, 71]. C57BL/6J mice show a characteristic and reproducible circadian rhythm to lights-on (zeitgeber time (ZT) 0–12, day time) and lights-off (ZT 12–24, night time) (Fig 5A and 5B; for a corresponding raw data plot see Fig 2 in S1 File). This pattern reproduced well across the four sites and confirm the previous observations [2]. The only minor deviation was the rather slow increase in activity among male mice following lights off at the IMG site.

Across the participating sites, the cumulative night time activity was about x3 that observed for day time and the diurnality appears to be different between sexes. In Fig 6, box plots show the metrics for the average day and night time activity and the relative effect of sex at the four sites (Fig 6a and 6b; see also Fig 3a, 3b in S1 File). Day time activity (Fig 6a and 6b) did not differ significantly between sexes (P = 0.481) but between facilities (p = 1.2E-34) and across cage-change cycles (p = 6.2E-10). During the active period with lights off there was a highly significant difference between males and females (p = 1.3E-16) and also some difference between sites (p = 1.1E-3) with a significant interaction between site and sex (p = 3E-6) (Fig 6a and 6b).

Consistent with this, the fraction of daily activity occurring during lights on, showed a significant difference between sexes (p = 4.7E-14) and sites (p = 1.4E-10) and also across cage-change cycles (p = 5.8E-20) (Fig 3a, 3b in S1 File).

## Spontaneous home-cage activity across a cage-change cycle by sex and site

A key rationale of this study is our previous observation that cage-change brings about a major alteration in spontaneous home-cage activity [2]. In order to compare the change of activity over days, irrespective of the total amount of activity, we normalised the activity of each day by the peak of activity during that cage change cycle.

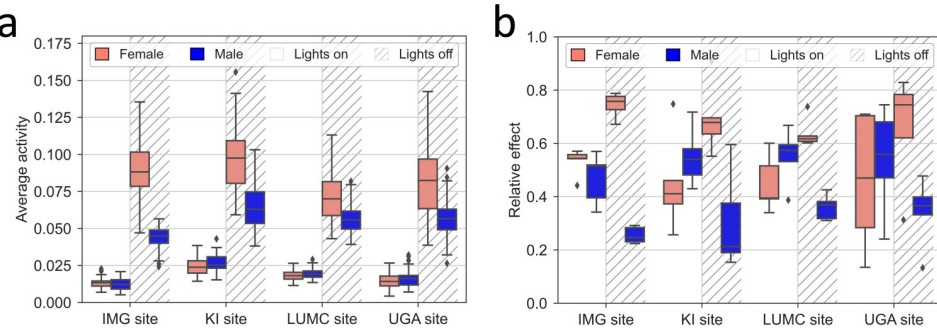

**Fig 6.** (a) Average activity during day and night time for male and female mice at the four sites. A) Boxplots (with median, 25% and 75% percentiles) show the average activity across the five cage-change cycles, for male (blue) and female (pink) mice at the four sites during lights on and lights off (grey stripes). Note the higher activity among female vs. male mice during lights off. (b) Box plots of the relative effect size of sex during lights on (clear background) and lights off (hatched background) for each site (see Material and methods). Male (blue) and female (pink). For raw data analysis see Fig 4 in S1 File.

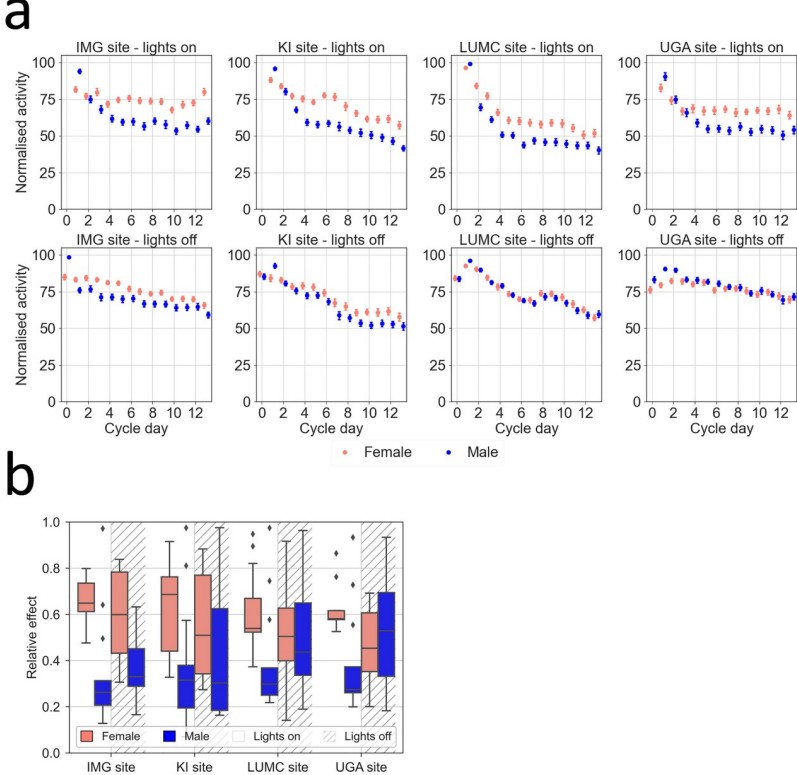

**Fig 7.** (a) Activity normalised to peak activity (%) during each day (upper row panels) and night (lower row panels) following a cage change. Both daily day and night time activity for each cage was normalised to the peak day/night activity of the cage over the respective cage change cycle. Average normalised activity ± SEM, for all male (blue) and female (pink) cages, for each day (abscissa) of the period across the five cage change cycles at each site (columns of panels). For day time, day 1 is the day that follows upon a cage change day while for night time day 0 is the night following the cage-change that took place during the preceding day time. Data was plotted separately by day and night for a better visualisation, but the light factor was tested as a repeated factor (subplot factor in nparLD testing). (b) Box plots of the relative effect size of sex during lights on (clear background) and lights off (hatched background) across cage-change cycles for each site (see Material and methods). Male (blue) and female (pink). For raw data analysis see Fig 4 in S1 File.

As evidenced by the data presented in Fig 7a, the highest in-cage activity was recorded immediately after cage change (at day 0–1 for lights off and day 1 for lights on; Fig 7) for both male and female mice followed by a gradual decrease in activity by 25%-50% (normalised data, p = 4E-129 to p = 6E-20; non-normalised data, p = 4.3E-14 and p = 7.3E-97; see Fig 3 in S1 File) and a difference between the day time and night time responses indicating that the most marked impact occurs during day time when the animals normally rest (Fig 7b; light as a sub-plot factor, normalised data, p = 5E-37 to p = 2E-2; non-normalised data p = 0 and p = 3.1E-87; see Fig 4 in S1 File). An effect by sex on this response was statistically significant only at the IMG and KI sites (Fig 7b; normalised data, p = 2E-8 and p = 6E-7, respectively; non-normalised data, p = 1.2E-4 and p = 3.2E-3, see Fig 4 in S1 File).

## Impact by housing density and bedding material on in-cage activity

Both cage housing density and the suitability of different types of bedding materials are currently subject of multiple studies (for references see Introduction). At the KI site, we compared in-cage activity using aspen chips (AC) with corn cob (CC) and different mouse holding densities (Fig 8; for experimental details see Material and methods).

Irrespective of the bedding material used and the housing density, the activity declined to the same extent and with a similar progression post cage change as in the multicenter study reported above (Fig 8a; p = 6.6E-69, 4 mice per cage; p = 7.4E-36, 2 mice per cage). There was a small but significant difference in activity decrease between housing on AC and CC with four mice per cage but not with two mice per cage (Fig 8a and 8b; p = 1.5E-4, 4 mice per cage; p = 0.41, 2 mice per cage).

## Use of cage-floor and position of latrine

The in-cage life of mice shows a clear organisation with a hierarchy of the standing of the individuals in the group, the building of nest(s) and the assignment of one or more areas as latrine (s). To a large extent, this cage life order is disturbed when we perform a cage-change.

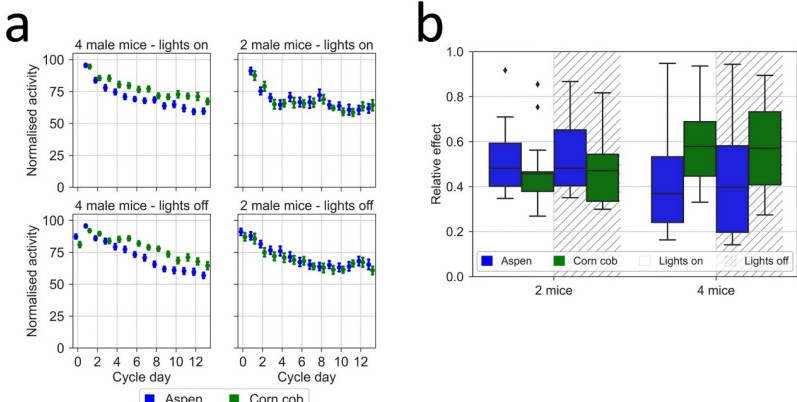

**Fig 8.** (a) Activity normalised to peak activity (%) during the light (upper panels) and dark (lower panels) periods, for n = 4 (left two panels) and n = 2 (right two panels) mice per cage, housed on aspen chips and corn cob (see the key to symbols). Data are average normalised activity values ± SEM. For day time, day 1 is the day that follows upon a cage change day while for night time day 0 is the night that follows upon the cage-change which took place during the preceding day time (day 0) (for a corresponding plot of non-normalised data see Fig 5 in S1 File). (b) Box plots of the relative effect size of type of bedding material on changes in activity during lights on (clear background) and lights off (hatches background) across the cage-change cycle for each housing density shown in (a) (see Material and methods). Aspen chips (blue) and corn cob (green).

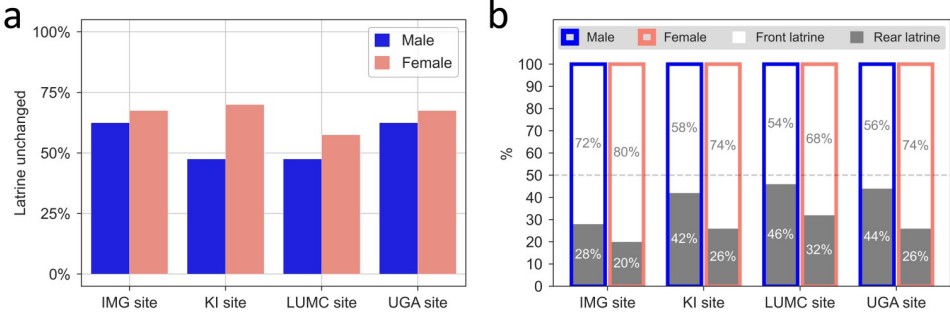

**Fig 9. Position of latrine(s).** a) Percentage of instances where latrine(s) position was not changed between two consecutive cage-changes by sex and site. b) Bar charts showing the percentage (all cages times cage-change cycles) of front and rear position of the latrine(s) in the cages with male (blue) and female (pink) mice at the four sites.

As described in Materials and methods (see details in, Fig 6 in S1 File), the positions of latrines were defined by the recorded drop in capacitance across the cage-change cycle due to the wetting of the bedding in and around the latrine. We found that the latrine predicted by DVC® agreed in 85% cases with the area with the highest ammonia concentration and that the placement of the latrine(s) varied between cage-cycles in all cages. The latrine was re-assigned to the same area of the cage in less than about 2/3$^{rd}$ of the cases, with males being more prone to shift localisation than females (Fig 9a). As depicted in Fig 9b, female mice tended to create the latrine(s) in the front of the cage, while male mice showed no clear preference for front vs rear position of latrines at the UGA, LUMC and KI sites. At the IMG site, males showed a similar preference as the females for having the latrine in the front.

The decision on the placement of the latrine(s) by the mice occurred shortly after cage-change. As shown in the heat map of the Frontality of activity (i.e. the percentage of activity that occurred in the frontal section of the cage floor) vs. latrine position across cage-change cycles (Fig 10 showing an example from the UGA site; examples from other sites are shown in Figs 7–9 in S1 File), male mice are more active in the regions of the cage floor being devoid of the latrine(s) already the day after a cage change. A trend that continued till the end of the cage-change cycle at each of the sites (see below).

Expanding on the in-cage activity (Figs 6–8 and 10) and latrine position data presented above (Fig 9), we deepened the analyses of Frontality (Fig 11) for each day of the cage-change cycle in relation to the locations of the latrines across all five cycles at each site. The data plotted for each site illustrate that the percentage of daily activity recorded from the frontal half of the cage-floor increases over time when the latrine is positioned in the rear, while it decreases when the latrine is in the front. This was significant at all sites (interaction between day and latrine position, p = 9E-34 to p = 1.2E-04).

Male and female C57BL/6J mice appear to differ in the evolution of Frontality over time, female mice tend to reduce activity by 5-to-10% in the latrine area and this occurs during the 2$^{nd}$ week of the cage change (lower panels) while males disclose a much more robust (~25%) shift of the activity to the latrine free area already on day one at the IMG, KI and UGA sites and on day two at the LUMC site (Fig 11a). This shift in activity was significant at the UGA and LUMC sites but not at the other sites (Fig 11b; interaction between sex, day and latrine position, p = 2.7E-02 and p = 1. 1E-02 respectively; while p = 4.4E-01 at IMG and p = 8.3E-02 at KI). Thus, latrine position had no statistically significant effect on female mice preference for frontality at the IMG and KI sites across the full cage-change cycle (Fig 11b).

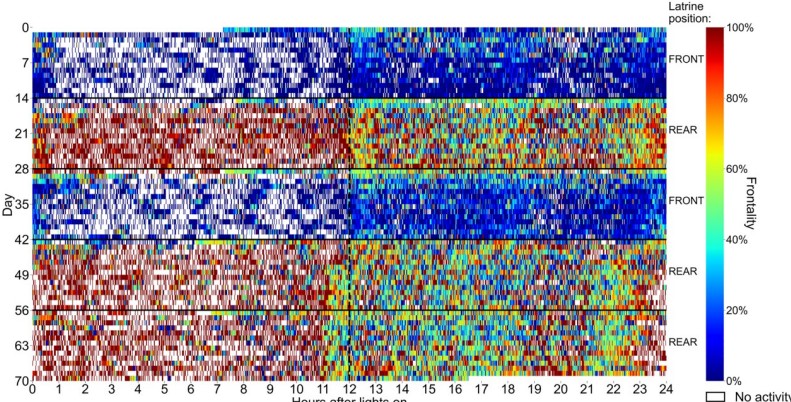

**Fig 10. Minute-based frontality and latrine positions day-by-day across cage-change cycles.** Heatmap showing the Frontality for each minute of one male example cage at the UGA site for the five cage-change cycles (for corresponding information of the three other sites see Figs 7–9 in S1 File). Frontality is the percentage of activity performed in the front-half of the cage-floor over the total activity performed on the entire cage floor, therefore red indicates that activity min$^{-1}$ is performed mostly in the front half of the cage, while blue indicates that activity is performed mostly in the rear. White denotes the absence of activity. Lights on at ZT 12 haves been indicated by a dashed black vertical lines so that the period with lights on is on the left part of the figure and the period with lights-off is on the right part. The position of the latrine(s) during each cycle is indicated at the right of the heatmap. Note the switch from summertime to wintertime week 47 (to 70).

In the separate experiment performed at the KI site, animals housed 4 to a cage tended to be less and less active in the region of the latrine (Fig 12; latrine factor, p = 2E-62; interaction between latrine and day, p = 2.8E-92), irrespective of the type of bedding materials used (bedding type factor, p = 0.61). In contrast, animals housed two per cage didn't shift their activity span resulting in no impact on frontality by latrine position irrespective of the type of bedding materials used (latrine factor, p = 0.32; bedding type factor, p = 0.11).

## Intra-cage NH3 levels across cage-change cycles

NH$_3$ levels (ppm) were measured repeatedly between successive cage changes across the front, the middle and the rear area of each cage floor under flow conditions at KI, LUMC and UGA (see Material and methods). Fig 13 shows the average NH$_3$ ppm calculated for each cage area (latrine area, and area opposite, which can be front or rear, and middle) at the end of the 14-days cycle, for each sex and site.

In Figs 10–12 we show that with days passing between cage changes the mice allotted more and more of their activity to the latrine free part of the floor. For this reason and also the observation that the latrine was never found to be in the middle portion of the cage, we deepened the analysis of NH$_3$ levels by analysing the NH$_3$ in the latrine area of the floor vs the opposite latrine free floor area (Fig 14). As evident from the plots, there is an increase in NH$_3$ ppm with more days post cage change (KI p = 6.6E-51, LUMC p = 2.2E-77, and UGA: p = 1.7E-90) with significant differences between the latrine area and the opposite latrine free area (KI p = 3.7E-6, LUMC p = 7.1E-11, and UGA: p = 5.1E-43) and between sexes (KI p = 4.1E-03, LUMC p = 1.8E-2, and UGA: p = 3E-8). In the latrine area, the NH$_3$ ppm remained at 0 on day 2 at both LUMC and UGA, and was slightly above 0 at KI. For males, it then increased and exceeded 25 ppm on day 5–6 and at KI and UGA exceeded 50–75 ppm, towards the end of the cycle (Fig 14). At LUMC, the progression was delayed during the first week, but then accelerated to ≥75 ppm towards the end of the cycle. In cages with females, ammonia levels in the

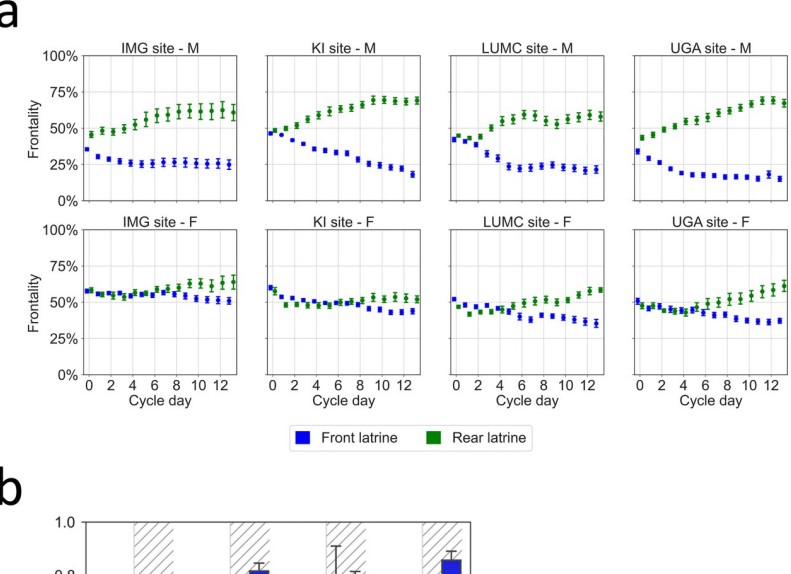

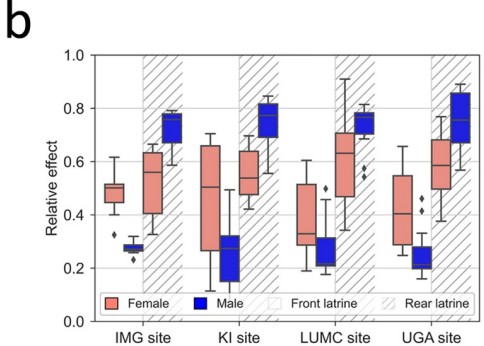

**Fig 11.** (a) Frontality of activity in relation to latrine position during the 14 days after cage-change. Plots show average ±SEM of daily (24 hours) Frontality across the five cage-change cycles, which is the percentage of daily activity performed in the front-half of the cage-floor over the total activity performed on the entire cage floor during the 24 hours of a day. Frontality is displayed for cages with the latrine in the front (blue line) or the rear (green line), for males (upper panels) and females (lower panels), for each IMG, KI, LUMC and UGA respectively. (b) Box plots of the relative effect size of sex on Frontality in the latrine and latrine-free area of the cage floor across the cage-change cycle for each site shown in (a). Males (blue) and female (pink); clear background frontal placement of latrine, hatched background with latrine in the rear.

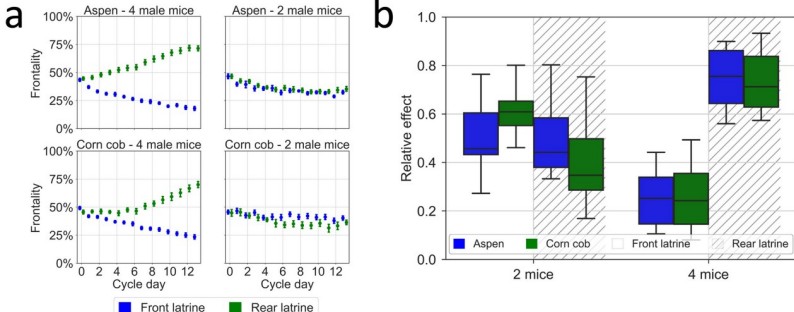

**Fig 12.** (a) Frontality of activity (%, ordinate) and position of latrine(s) days after cage-change (abscissa) with different bedding material (aspen chips top panels, corn cob lower panels) and four (left two panels) vs two (right two panels) male mice per cage at KI. Plots showing average ±SEM of daily (24 hours) Frontality, which is the percentage of daily activity performed in the front-half of the cage-floor over the total activity performed on the entire cage floor during the 24 hours of a day. Frontality is displayed for cages with the latrine in the front (blue symbols) or the rear (green symbols). (b) Box plots of the relative effect size (see Material and methods) of bedding material with frontal (clear background) and rear (hatched background) latrine position across the cage-change cycle for each housing density shown in (a). Aspen chips (blue) and corn cob (green).

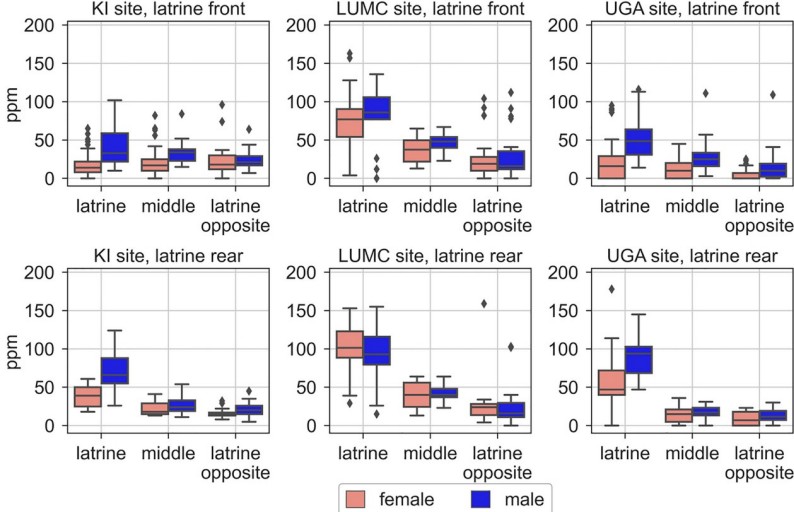

**Fig 13. Ammonia (NH₃) measurement (ppm) on the last day of a cage-change cycle in the front, middle and rear of the cage, for each site and sex.** Box plots with median, 25% and 75% quartiles of the three floor areas across cage-change cycles for the three sites (column of panels) separated by sex (see colour code key for sex).

latrine zone at KI and UGA remained below or touched 50 ppm across the whole cycle while during the second week NH₃ ppm increased rapidly to reach the same end values as for males at LUMC. The floor area opposite the latrine showed only small differences between sexes, progressed more slowly and remained below or was about 25 ppm as end values at all three sites. As expected based on the ventilation dynamics of the IVC cages, the ammonia levels across the cage-change cycle were higher in the latrine area when this was located in the rear compared to the front area (Fig 14). There was a correlation between NH₃ ppm measured in

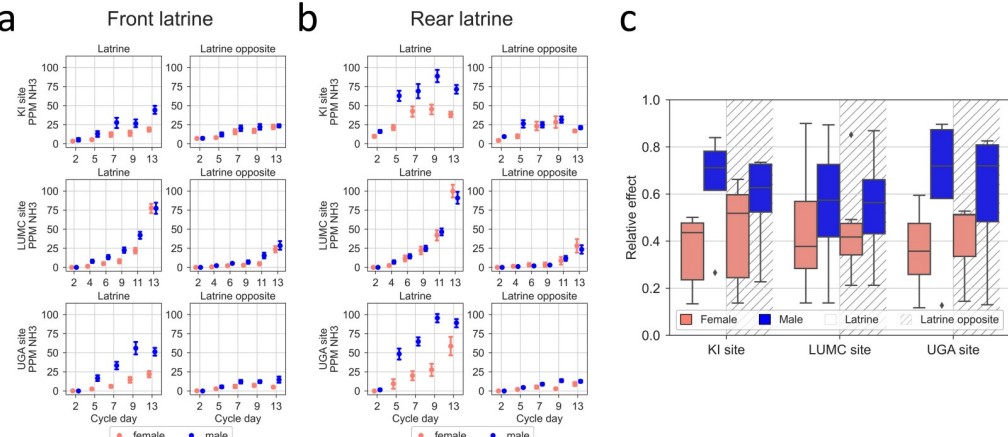

**Fig 14.** (a) NH₃ measurements in relation to latrine position across days after cage-change. Plots show the average ±SEM of NH3 ppm, measured over the latrine and the opposite area for when the latrine was in the front (left six panels) and when the latrine was in the rear (right six panels), for males (blue line) and females (pink line) for KI (top four panels), LUMC (middle four panels), and UGA (bottom four panels) respectively (b) Boxplots showing the relative effect of sex (males in blue, females in pink) on recorded ppm NH₃ in the latrine and latrine-free cage-floor area (latrine area = clear; latrine-free area = hatched), respectively, across the cage-change cycle at each site. Males in blue and females in pink.

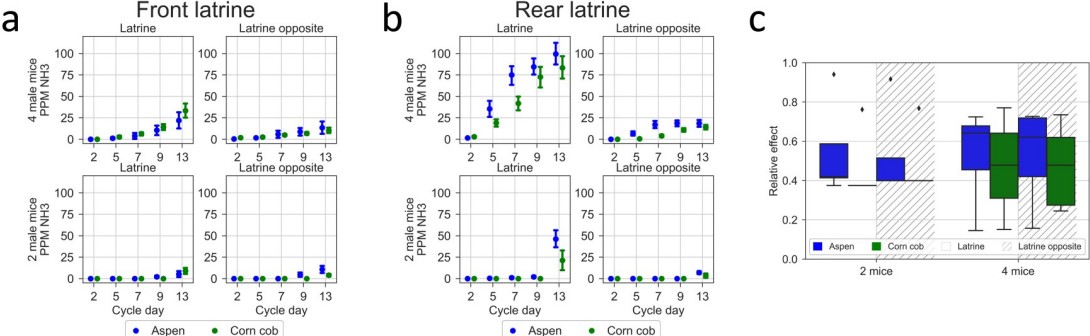

**Fig 15. NH₃ measurements in relation to latrine position on different days post cage-change with aspen chips and corn cob, and either 4 or 2 animals per cage at KI.** Plots show the average ±SEM of NH₃ ppm in the latrine and opposite area of the cage floor for latrine in the front (left four panels) and latrine in the rear (right four panels) of cages with mice on either corn cob (green) or aspen chips (blue) bedding and with a housing density of four (upper two panels) or two (lower panels) animals per cage at the KI site. (b) Boxplots showing the relative effect of bedding material (aspen chips in blue, corn cob in green) on recorded ppm NH₃ across the cage-change cycle in the latrine and latrine-free (latrine area = clear; latrine-free area = hatched) cage-floor areas, respectively, at the housing density of two and four mice.

the latrine area and in the opposite latrine free area at the three sites (Spearman rank correlation; cages males $r_s$ = 0.72, p = 1.8E—17 down to $r_s$ = 0.48, p = 3.7E-9, when the latrine is in the rear; $r_s$ = 0.87, p = 2.2E-44 down to $r_s$ = 0.31, 5.9E-5 when the latrine is in the front; female cages $r_s$ = 0.82, p = 2.4E-16 down to $r_s$ = 0.62, 1E-11 when the latrine is in the rear, $r_s$ = 0.82, p = 8E-45 down to $r_s$ = 0.59, p = 1.1E-20 when the latrine is in the front) suggesting a gradient of NH₃ ppm across the cage floor (see also Fig 13 and [42]).

## Housing density, latrine position and bedding material impact on in-cage activity and NH3 levels

At the KI site, animals were also kept on CC and housed at a reduced density (Fig 15a and 15b). With four animals, the small difference in average NH₃ ppm values between beddings was not statistically significant (15b, p = 0.47). The ammonia levels increased with both beddings types (Fig 15a; p = 2.8E-35) as time progressed post-cage-change and there was a significant difference between the latrine area and the opposite latrine free area (p = 1.2E-8). These results are consistent with the multicenter study reported above. With a housing density of two animals per cage, the levels of NH₃ ppm remained very low throughout the cage-change cycle albeit with a small difference (p = 1E-4) between AC and CC (Fig 15a and 15b). Also in these sets of cages, there was a significant covariation of NH₃ ppm measured in the latrine and latrine-free cage-floor area (Spearman rank correlation, four mice on CC r = 0.85, p = 5.2E⁻³¹ and = 0.60, p = 4.3E⁻¹⁸, with latrine in the rear and in the front, respectively; four mice on AC r = 0.90, p = 6.8E⁻⁵³ and r = 0.71, p = 6.1E⁻²²; two mice per cage on CC r = 0.90, p = 8.3E⁻³⁹ and r = 0.74, p = 4E⁻²⁶; two mice on AC r = 0.84, 2.2E⁻²³ and r = 0.75, p = 1.2E⁻¹⁵).

## Number of animals, biomass and in-cage NH₃ ppm

A direct comparison of NH₃ ppm in cages with male mice and that in cages with female mice at all three sites revealed a statistically significant difference with higher NH₃ ppm in cages with male compared to those with female mice (Fig 16). Since the excretion of urea, a key component of nitrogen homeostasis, is to some degree proportional to the biomass of the organism, we used the collective body mass of each cage to compute average NH₃ ppm per gram

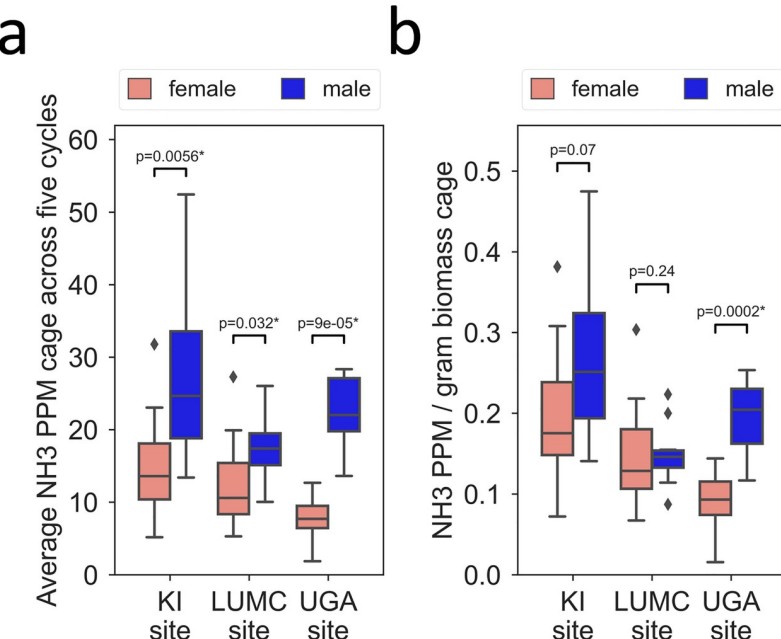

**Fig 16. Boxplots of average $NH_3$ ppm in cages with four female or four male mice (see key for male and female) across five cage changing cycles at KI, LUMC and UGA.** Left panel (a): A direct comparison of $NH_3$ ppm in cages with male mice and that in cages with female mice (left panel). Effect size by sex was large at each site ($CL_{EE}$: 0.84, 0.75 and 1.0, respectively; see material and method). Right panel (b): $NH_3$ ppm per gram biomass in cages with male mice and in cages with female mice (p values for Mann-Whitney U-test have been indicated). With $NH_3$ ppm normalized to in-cage biomass, the effect by sex was still high ($CL_{EE}$: 0.98) and statistically significant at UGA but lower and did not reach statistical significance at KI ($CL_{EE}$: 0.7) and LUMC ($CL_{EE}$: 0.6).

biomass. We then found that the difference in $NH_3$ level between sexes across cycles did not reach significance at KI and LUMC but remained significant at the UGA site (Fig 16b).

## Vertical slot position, light level, $NH_3$ ppm, and in-cage activity

As detailed in the Materials and methods section, we rotated each cage at every cage change to mitigate an impact by light level or potential non-uniformity of the ventilation. Post hoc analyses of the activity and $NH_3$ ppm data collected from each slot (covering ten different cages each inserted for two weeks) revealed no gradient of either activity or $NH_3$ ppm with slot position (see Figs 10–14 in S1 File).

## Intra-cage bacterial growth and $NH_3$ levels during bi-weekly and weekly cage-changes, and post hoc histopathological examination of the upper airways

At the KI site, bacterial load (CFU) inside the cages was assessed (n = 20) through swabbing at the end of the five cycles of bi-weekly cage-change (see Material and methods). The cage-change routine was then shifted to weekly cage change for two rounds and at the end of the second round all cages were swabbed again for assessment of CFUs. Pair-wise analysis of bacterial load with bi-weekly and weekly cage changes, respectively, revealed only small difference in CFU between the cage change intervals (p = 0.095; Fig 17).

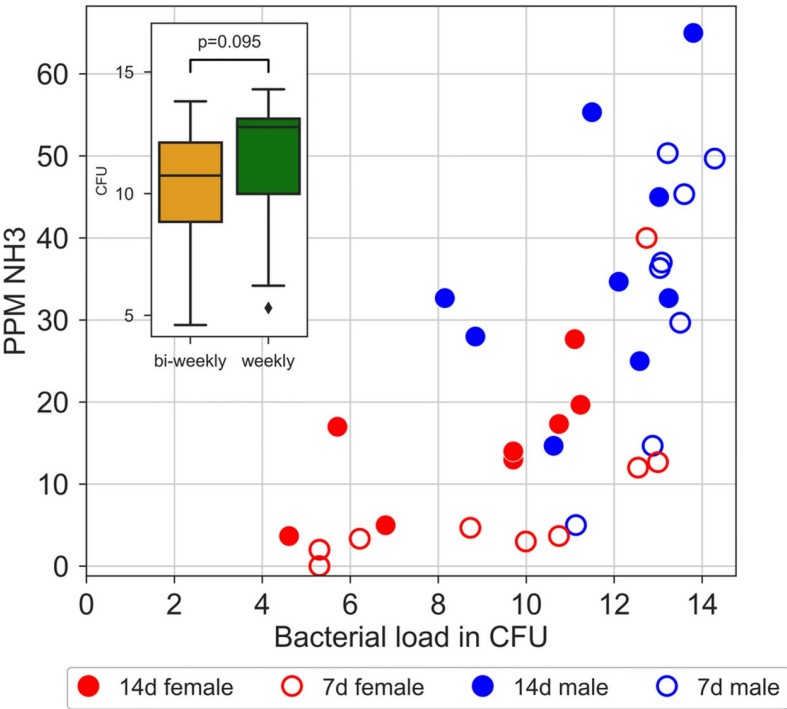

**Fig 17. Bacterial growth load (CFUs) and average of NH₃ ppm across the cage-change cycle in cages changed bi-weekly (filled circles) vs. weekly (open circles).** Data from male and female cages are colour coded, see key in diagram. Inserted in the diagram is a box plot of bacterial load for both samples and the obtained p-value for the statistical pair-test (Wilcoxon matched pair).

Measurements of NH₃ ppm was continued during the two weekly cage-changes to allow us to assess covariation between CFU and in-cage NH₃ ppm. In both the weekly and bi-weekly data sets of NH₃ ppm and CFUs there was a significant correlation between the two metrics (Spearman rank correlation, weekly: r = 0.88; bi-weekly: r = 0.76; Fig 17, see also Fig 15 in S1 File).

At the end of the five cycles of bi-weekly cage changes and the two final cycles with weekly cage-change, one animal per cage was randomly selected for histopathological analysis [13, 24, 57] of the upper airways to assess acute and cumulative impacts by in-cage bacterial load and ammonia (SPF group; Fig 18a). Mice of the same strain, sex and age but bred and housed under germ-free (axenic) conditions served as reference group with no lifetime exposure to NH₃ (GF group, Fig 18a). The tissue section analyses were conducted blinded on coded slides by two histopathologists ranking (0–5) the presence (>0) and extent (1–5) of each of the seven parameters (Fig 18a). The concordance between assessors was high (>90%; Fig 18a). In addition, the number of Caspase-3 positive cells mm⁻¹ was counted in the respiratory (RE) and olfactory (OE) epithelium, respectively. There were no signs of tissue necrosis or scarring while seven of the nine morphological features assessed (Table 2) were encountered in both SPF and GF animals. Inflammatory cell infiltrate in the vomeronasal organ was more prevalent in GF mice while hyalinosis in both the olfactory and respiratory epithelium, and Caspase-3 + cells in the RE were more frequent in SPF animals. The remaining five histological hallmarks present in both groups showed no difference between groups (Fig 18b). Thus all-in-all 4 parameters were either exclusively present in SPF animals or at least significantly more

a

| Region | Observation | GF | n=20 | SPF | n=20 | MannW U | | CL |
|---|---|---|---|---|---|---|---|---|
| | | median | range | median | range | p | n | Effect size |
| Nasal septum | NS Accumulation of "amyloid" | 2 | 1---3 | 2 | --- | 0.1068 | 40 | 0.60 |
| Respiratory epithelium (RE)(septum, turbinates) | RE hyalinosis | 0 | 0---2 | 1 | 0---3 | 0.0047 | 40 | 0.74 |
| | RE goblet (mucous) cell hyperplasia | 0 | 0---2 | 0 | 0---2 | 0.4196 | 40 | 0.57 |
| | RE caspase3+ cells* | 0.03 | ±0.04 | 0.75 | ±0.76 | <0.0001 | 39 | 0.88 |
| | RE inflammatory cell infiltrate | 0 | 0---2 | 1 | 0---2 | 0.0927 | 40 | 0.64 |
| Olfactory epithelium (OE)(dorsal meatus) | OE hyalinosis | 0 | --- | 0 | 0---3 | 0.0012 | 40 | 0.73 |
| | OE caspase3+ cells* | 0.75 | ±0.27 | 0.94 | ±0.69 | 0.687015 | 39 | 0.54 |
| | OE inflammatory cell infiltrate | 0 | --- | 0 | 0---1 | <0.0001 | 40 | 0.95** |
| Other | Nasal gland cysts | 0 | 0---2 | 0 | 0---2 | 0.4696 | 40 | 0.55 |

*Each observation was ranked by two specialists from 0 to 5; with 0 -not present, 3 -medium engagment and 5 - severe engagement)*

Agreement between assessors: 98%±0.03

Agreement between assessors: 93%±0.08

\*mean±SD mm$^{-1}$

\*\*Not observed in GF, large number of ties

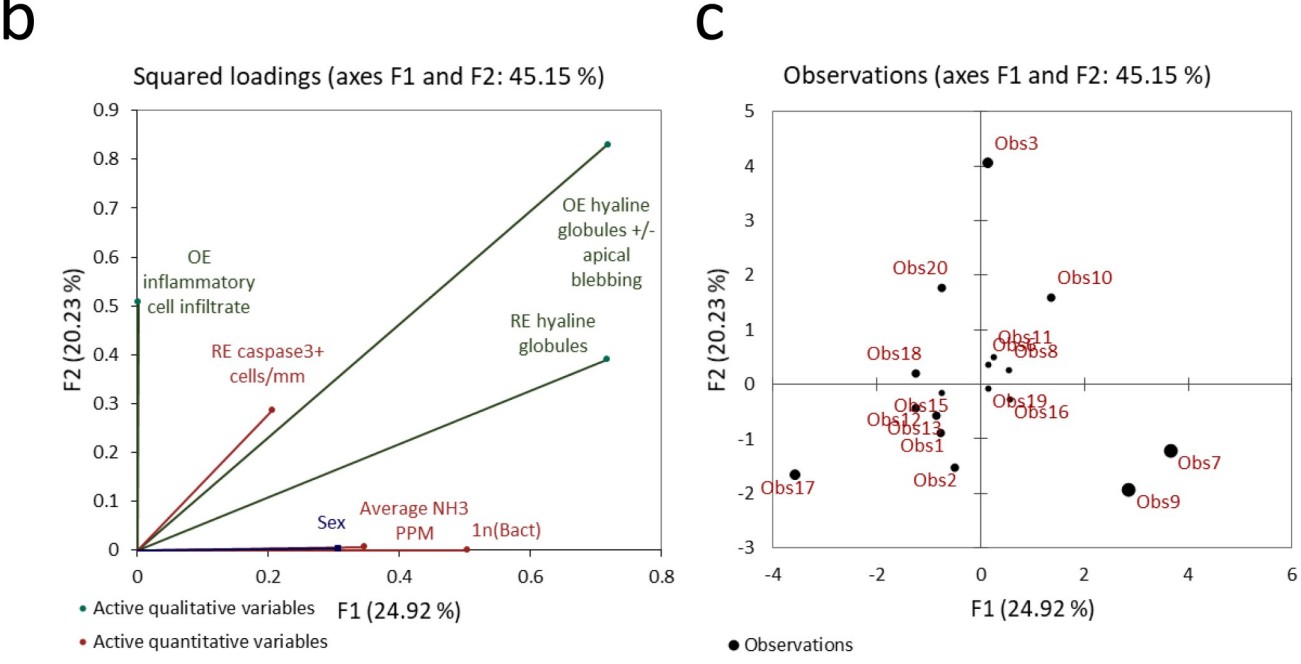

b

**Squared loadings (axes F1 and F2: 45.15 %)**

F2 (20.23 %) vs F1 (24.92 %)

OE inflammatory cell infiltrate

RE caspase3+ cells/mm

OE hyaline globules +/- apical blebbing

RE hyaline globules

Sex   Average NH3 PPM   1n(Bact)

• Active qualitative variables
• Active quantitative variables
▪ Supplementary qualitative variables

c

**Observations (axes F1 and F2: 45.15 %)**

F2 (20.23 %) vs F1 (24.92 %)

Obs3, Obs20, Obs10, Obs18, Obs11, Obs5, Obs8, Obs15, Obs12, Obs13, Obs19, Obs16, Obs1, Obs17, Obs2, Obs7, Obs9

● Observations

**Fig 18.** (a) is a tabulation of the ranks for the nine included variables (see Material and methods) with univariate statistics (Mann-Whitney U) of differences between the GF and the SPF groups and the common language (CL) effect size of the difference between groups. (b) shows the loading of qualitative and quantitative, and supplementary qualitative variables on F1 and F2 in the mixed PCA. The SPF mice separated by sex on F1. (c) shows the contribution by each observation, numbers ≤10 are males while >10 are females.

prevalent in this group. These four histological hallmarks (qualitative variables) were included in a mixed model PCA to assess covariation with CFUs and $NH_3$ levels (quantitative variables) while controlling for sex (Fig 18c). The covariation of qualitative and quantitative variables was low (F1 plus F2 ~45%). $NH_3$ ppm and CFUs contributed mainly to F1 while inflammatory cell infiltrate in the OE contributed to F2. SPF animals were well-separated by sex along F1 (Fig 18b and 18c).

## Discussion

Here we report the results of a multicentre study at four research institutes in Europe conducted in 2018–2020. The rationale was to shed light on the in-cage activity of a commonly used laboratory mouse strain across a cage change cycle of two weeks and to what extent in-cage levels of $NH_3$ impacted animal activity, dynamics of the use of cage floor space relative to the location of the latrine, and health of the mice. Another objective was to analyse to what extent the data reproduced across facility sites. This is the first report of a study involving the measurement of in cage $NH_3$ concentrations, where in each cage measurements were taken at three positions across the cage floor covering the entire width of the cage at all three positions under IVC flow conditions. In-cage ammonia concentrations were measured at different time points between cage changes using a commercially available portable electrochemical sensor device [41]. Male and female mice of three different C57BL/6J substrains were housed in GM500 IVC cages. Cages were changed every other week. Spontaneous activity on the cage-floor was recorded 24/7 with DVC$^®$ technology [2]. Novel to this study is that we used the DVC$^®$ electrode capacitance readings to decide on the location of in-cage latrine(s), enabling us to relate both $NH_3$ ppm and activity to the position of the latrine(s). At the KI site, bacterial load was measured and the airway histology of the SPF mice was compared to that of a separate cohort of germ free C57BL/6J mice. At none of the sites, animals were lost or had to be removed from the study prematurely. At the four sites, all animals exhibited weight increases similar to what is considered normal for C57BL/6J mice during the study period of 10 weeks and the additional 2 weeks at the KI site. In addition, the effects of reduced cage occupancy, two versus four animals, and different bedding types, corncob versus aspen chip, on the increase in $NH_3$ concentrations and on the spontaneous activity of the animals were investigated at the KI site only.

Although housing conditions were standardised as much as possible there were differences between sites with respect to the timing of lights on and lights off and the presence of a dawn and dusk period. At two centres, animals were handled with forceps while the two other centres used tail handling. Obviously, the caretakers differed between sites. It has been reported that the sex of the experimenter can affect apparent baseline responses in behavioural testing [72]. At the LUMC, all staff members involved in the experiments were male, while at the other sites the team consisted of male and female caretakers. Since all animals were housed in IVC cages and cages were only opened under laminar flow conditions in downflow cage changing stations, the effect of the sex of the experimenter on experimental outcomes is thought to be minimal.

### In-cage activity and impact by position of latrine(s)

Our cumulative records show a distinct circadian rhythmicity of in-cage activity. The data replicated well across sites and confirm previous observations [2]. The only notable difference was the rather slow onset of activity increase following lights off at the IMG site. This deviation could not be explained by light off being sudden or through a passage in dusk since two sites had no dawn and dusk period and two sites did of 10 and 30 minutes respectively. Also the

timing of the lights on and off could not explain the difference, since both IMG and KI had set lights off at 4:00 pm and lights on at 4:00 am. The onset of activity increase following lights off at KI was rather similar to the other two centres (UGA and LUMC) with lights off set at 7:30pm.

Our results confirm that female C57BL/6J mice are more active than age-matched male mice [2]. However, we found that the higher female activity appeared to be constrained to the dark phase. Hence, the difference between day and night time activity was greater in female than in male mice. In response to a cage change, male and female mice exhibited an instant increased of activity during both day and night time. The enhanced activity decreased during the first week at all participating sites and continued to decline at slower pace during the second week. This is in line with an earlier study with female C57BL/6J mice with weekly cage changes [2]. The amplitude of the increase after cage change and the following drop in activity varied between 25 and 50% during the first week depending on the site and the sex, levelling off during the second week (~-10%). Similar results were found during the extended experiment at KI with different bedding types and only two animals per cage, showing that this reaction following a cage change event is extremely robust across sites, sexes, animal housing density and independent from bedding types. In relative terms, the impact of the cage change on in-cage activity was more marked during day time when the animals rest than during lights off when the animals are active. Our findings are in line with others reporting altered behaviours and responses, and disrupted sleep patterns [1, 9, 45–49, 73].

An alternative approach to a predetermined fixed time interval between cage changes is the cage change on demand or 'spot cleaning'. Under that regime, the technician decides when a cage needs changing most often based on an SOP with images representing different degrees of soiling. Our results confirm that a cage change impacts the activity and sleep patterns hence are likely to impact scientific results. It should, therefore, be considered an experimental variable and included when designing the experiment. This is easier with a fixed cage change frequency than with cage change on demand. A bi-weekly cage change is then preferred over a weekly cage change.

We used the drop in electrode capacitance that occurs with increasing water content of the bedding to estimate the location of the latrine(s) across the twelve electrodes covering the floor area [52]. Considering all sites, the latrine position identified by the DVC technology agreed in 85% of the cases with the area with the highest $NH_3$ concentrations measured. In the remaining 15% of the cases, $NH_3$ values were extremely low and comparable between front and rear positions and therefore less informative to assigning the actual latrine position. In all cases, the latrine position was determined by visual inspection as well.

Our data show that even though food and water were presented more towards the rear of the DVC® GM500 cage type used and the rear of the cage was the more shaded area of the cage, male mice showed no clear preference for creating the latrine in the front or rear of the cage while female mice showed a preference for the front of the cage. Only at IMG, both male and female mice had a preference for the front of the cage inspite of the higher light intensity measured across the front of the rack compared to the other centres. This finding could not be linked with a causing factor except that that rack was placed against a wall bordering a corridor albeit with low traffic intensity and no detectable transmission of sound in the audible range for humans.

Depending on the location of the latrine, mainly male mice exhibited increasingly more activity at the opposite site of the cage away from the latrine as the time between cage changes progressed. More in detail, independent from the latrine position, immediately after cage change (day 0) males are already more active in the area opposite from the latrine and this will become more pronounced during, in particular, the first week but progresses at a slower pace

during the second week at two (KI and UGA) of our fours sites. Still at the end of the two weeks, 20%-25% of the daily activity across all sites is in the latrine part of the cage floor. Females show a different pattern of activity with no change in preference for the two floor areas during the first week, followed by a tendency (5–10%) to allocate more of the daily activity in the latrine free area. A tendency that became statistically significant at the LUMC and UGA sites only. These patterns are highly reproducible between sites also in the extended KI experiment with the two bedding types. However, this shift in activity span of male mice disappeared entirely when the cage occupancy was reduced to two animals per cage, implying that housing density can be a significant factor.

Whether the shift in activity span is accompanied by an increase in anxiety or stress levels will have to be confirmed in a subsequent study. The European Directive, 2010/63/EU, prescribes 330 cm$^2$ as the minimum enclosure size with defined floor area dimensions per animal depending on their body weight [74]. These minimum enclosure dimensions were set after the recommendations of the Council's Group of Experts on Rodents and Rabbits. The expert group concluded that figures for minimum cage sizes and maximum stocking densities can never be scientifically 'proven' for reason that the crucial point is the interaction between space, the structure of the cage, the animals and the type and quantity of enrichments provided [75] (see also [76]. In operant conditioning experiments, female C57BL/6 mice continued to exit the enriched cage and enter the empty one at all the costs imposed, but did not discriminate between the amounts of additional space offered [77, 78]. The author argued that the motivation to exit the enriched cage and enter the empty cage was due to monitoring, patrolling or information gathering, independent of any attraction or aversion to either of the cages. Based on these findings Gaskill et al. suggest that when considering space offered by larger cages, this may mean that mice will patrol the perimeter of the larger cage, but spend most of their time near the nest [79]. Determining the optimal combination of available floorspace and the floorspace area 'agreeable' to mice should be subject of future studies. These studies should incorporate measurements of physical and mental wellbeing parameters and consideration of the observed difference between sexes.

## Ammonia (NH$_3$)

A likely explanation for the shift of activity away from the latrine is that the animals avoid the areas of the cage that are becoming soiled with increasing concentrations of chemicals associated with the latrine [15, 19, 26]. The reduction in housing density from four to two animals lead to lower NH$_3$ concentrations and less soiling suggesting that NH$_3$ may drive this behavioural response. However, controlled NH$_3$ exposure studies indicate that levels up to 100 ppm are not aversive (preference test with clamped chamber) to mice [32]. In a study of voluntary wheel running (Swiss strain) Tepper et al. [80] found no consistent impact on activity by exposing the mice to 100 ppm NH$_3$ for 2 days while exposure to 300 ppm NH$_3$ substantially reduced the spontaneous activity.

Consistent with studies examining both sexes [7, 21, 81], the measured levels of NH$_3$ were significantly higher in cages with male than with female mice (Fig 16a). A difference that can be explained in part by the larger biomass of the males (Fig 16b). Our conclusion is that strain information, number of animals and sex need to be complemented with biomass in order to safely control in-cage NH$_3$ levels. Notably, the difference in NH$_3$ levels between sexes was larger in the latrine area than in the latrine free area. In line with previous studies [13, 21], the increase in NH$_3$ showed some variability between cycles and within cycles. Variances within series of measurements of NH$_3$ may be generated by the in-cage activity preceding the

measurement, for example timing of the use of the latrine and activity at the latrine(s) which both may facilitate release of $NH_3$ from soaked bedding.

Since we measured $NH_3$ at the front, middle and rear of each cage, we could clearly demonstrate an $NH_3$ gradient with higher concentrations over the latrine area and lower $NH_3$ concentrations detected over the adjacent middle and opposing areas of the cage floor (see also [10, 16]). Overall, $NH_3$ concentrations in the front of the cage were lower than those in the rear of the cage irrespective of sex, housing density or bedding type. Possible explanations for this finding are the direction of the airflow from the rear top of the cage towards the front then down and back across part of the cage floor up to the rear top, and the spill of diet under the hopper situated closer to the rear of the cage creating favourable conditions for urease producing bacteria to grow (see also [10]).

For female mice the level of $NH_3$ across the five cage-change cycles were at, or below, 25 ppm in the latrine free area of the cage irrespective whether the latrine was in the front of the cage, upstream, or in the rear of the cage, downstream from the airflow. In the latrine area, the $NH_3$ concentration remained below 25 ppm until day 12 of the cage change cycle at the three sites where $NH_3$ was measured (KI, LUMC, and UGA).

With males, $NH_3$ exceeded 25 ppm on the final day before the cage change in the latrine free area while ammonia levels exceeded 25 ppm during the first week at KI and UGA but not until the middle of the second week at the LUMC. While $NH_3$ ppm continued to increase at the LUMC (up to 75 ppm) until the final day of the cage-change cycle, the corresponding progression levelled off at the two other sites (end values at ~50 ppm). This may be explained by a different dynamics of in-cage bacterial growth between sites as discussed below.

As demonstrated by Rosenbaum and co-workers [11] (see also [9, 15, 16, 28]) varying the air change rate or the amount of the bedding material can modify in-cage $NH_3$ levels. Here we used the seventy-five air changes per hour (ACH) and the amount of bedding material recommended by the vendor to optimise in-cage conditions (see Material and methods). Lower rates of ACH may compromise the in-cage environment with higher holding densities (> two animals) but higher ventilation rates may also have disadvantages since air turbulence and a strong laminar flow may compromise the wellbeing of the mice [8]. Rather than changing the ACH, choosing the right type and amount of bedding material is a better way to optimise the in cage microenvironment (see also below). Tateryn et al. reported corn cob to show better $NH_3$ control than aspen chips already as early as 4 days after cage changing and throughout the 2-week measurement period [31]. Using a housing density of either 4 or 2 male mice, we could, however, not replicate these observations in the current study suggesting that this issue deserves further investigation.

Given the choice, data in the literature indicates that mice prefer aspen chips over corn cob [82]. However, consideration should also be given to the fact that both wood chips and corn cob are complex organic composites that contain a range of chemicals potentially posing a hazard to the animals' health and/or interference with experiments on the animals. In particular, corn cob and pine chips have been considered toxic or troublesome in a variety of contexts [83–87], but also other wood shavings like the widely used aspen chips have only been tested for a non-exhaustive library of chemicals and the users are left uninformed about potential interference or toxic components of this bedding. A leap forward in improving housing standards for small rodents would be to introduce a synthetic bedding material with a texture and composition meeting the animal's need, is tolerated well and reduces the risks for aversive chemical side-effects at the same time, including exposure to $NH_3$. To our best knowledge there is only one such effort published to date [29].

Our results show that male mice decrease the amount of activity in the latrine zone of the cage floor already before $NH_3$ levels increased, indicating that other urine associated chemicals

and odours affect their behaviour. C57BL/6J is a highly inbred strain resulting in different male mice possessing the same chemo-signals in their urine associated with MUP and MHC peptides [88, 89]. This should act to prevent C57BL/6J males from discriminating odours of male mice of the same inbred strain, regardless of their age. Thus, it seems unlikely to be chemicals associated with the marking of territorial boundaries that repelled male mice from being active in the latrine zone. In contrast, C57BL/6J males did display discriminative deposition of urine marks toward outbred CD-1 males and toward adult C57BL/6 females [90]. Also, single housed C57BL/6J male mice marked more than pair-housed males. Hence, this issue certainly deserves further attention.

## Bacterial load and ammonia levels

The low level of $NH_3$ across the first few days post cage-change is the results of the process by which $NH_3$ is generated and that we used autoclaved cage materials including bedding and irradiated food, thus the conditions in a new cage were close to sterile. The production of $NH_3$ from urea by bacteria-derived urease escalates as bacterial load increases overtime. The origin of these bacteria is the microbiota of the SPF animals and the accidental contamination with agents, like *Staphylococcus aureus*, by staff. This was also the rational to include germ free mice of the same strain and age as a no NH3 exposure reference group in this study [53–55]. Our data on bacterial load (CFU) are in agreement with previously published data [17]. Notably, the difference in CFUs between cages subjected to weekly and biweekly cage-change was small indicating that the colonisation became saturated towards the end of the first week. This could explain the levelling off of in-cage $NH_3$ build-up during the second week post cage change at KI and UGA. The slower increase of $NH_3$ ppm recorded at LUMC, may have been caused by a more protracted bacterial colonisation during the first days after cage change which caught up during the second week to generate $NH_3$ en-par with the other sites towards the end of the cycle. Differences between sites may well be caused by differences in microbiota with animals coming from different vendors. All three facilities maintained an SPF status considering the FELASA recommended list of agents to be tested for as exclusion list [5].

In both the bi-weekly and weekly data set there was a significant correlation between CFU and $NH_3$ concentration. However, while low levels of in cage $NH_3$ ($<25$ ppm) covary closely with CFUs, high ppm values (31–65 ppm) were more scattered especially in cages changed bi-weekly. The poorer covariation between high levels of $NH_3$ ($>25$ ppm) and CFUs may be explained by the shifting towards urease producing bacterial strains at the expense of other bacterial species as time progresses. Another not mutually exclusive explanation is that $NH_3$ being a sticky molecule may accumulate in the organic bedding material until the bedding becomes saturated after which the concentration of the gaseous $NH_3$ will increase.

## Histopathology

$NH_3$ has attracted considerable attention as a significant micro-environmental factor in housing of small rodents because it may induce lesions in the upper airways compromising health and it is a potent irritant of sensory nerve endings of the nose and the eye [7, 9, 13, 16, 20, 21, 24, 25, 35–38, 40]. The levels of bacterial CFUs and in-cage $NH_3$ ppm recorded in this study induced no overt impact on the health of the animals. However, a number of studies have reported histopathological findings in the nasal cavity and/or lung following one to several weeks of in-cage $NH_3$ levels corresponding to those measured in our study [13, 20, 91]. Therefore, at the end of the last cage-change cycle, 10 male and 10 female animals were randomly selected from each cage at the KI site and their upper airways dissected for histopathological analysis (see Material and methods). These SPF mice were compared with a group of mice

raised and kept under axenic conditions (GF). No overt signs of tissue necrosis or scarring were observed in either group. In general the scoring of histopathological parameters (see Table 2 in Material & methods) was low and never exceeded three on a scale 0–5 in both groups. Where signs of tissue alterations were present in both SPF and GF mice (7 of 9 parameters), these may be driven by other environmental factors such as dust or chemicals deriving from the bedding material rather than exposure to bacteria or $NH_3$. The presence of hyalinosis and apoptotic cells (Caspase-3+) in the RE of SPF mice were significantly more prevalent than in the GF mice. Furthermore, hyalinosis and inflammatory cell infiltrates were exclusively encountered in the OE of the SPF animals. Inspite the significant difference between the observations made in SPF and GF mice, these parameters showed no, or a very low covariation (mixPCA) with $NH_3$ ppm and bacterial load (CFU) in the SPF group. The absence of marked histopathological alterations in the upper airways and the low degree of co-variation with $NH_3$ level and bacterial load is not surprising but rather expected against the results published [32, 33, 38, 40, 92–94] where exposure levels were well-controlled showing that weeks of exposure to >56 ppm $NH_3$ are needed to induce tissue alterations to the upper airways in mice and rats. Furthermore, our results are in line with several studies assessing histopathological changes and in-cage measurement of $NH_3$ levels (<100 ppm) under different mouse housing paradigms [7, 11, 14, 16, 21, 24, 42]. Given the results from controlled exposure studies (see above), the explanation of the discrepancy found in the literature (see above) concerning histopathological airway alterations in small rodents housed under similar in-cage $NH_3$ ppm (<56) may be due to measured $NH_3$ levels not corresponding closely to the individual animal's exposure to $NH_3$ and, in addition, that differences in $NH_3$ measuring technique used between studies may preclude closer comparisons of the results [41]. Another not mutually exclusive explanation is that histopathological changes to the upper airways were generated by other environmental chemicals and/or dust, not being $NH_3$. This may then explain that seven of the nine parameters examined were present also in the GF mice. Bolon et al. [33], compared nose histopathology after controlled exposure to 0 and 300 ppm $NH_3$ during 6 hours per day over 5 or 10 days, and observed that high $NH_3$ ppm was associated with anterior nasal lesions but not lesions of the olfactory epithelium (see [40]). This is consistent with the lack of correlation between observed in-cage $NH_3$ ppm and inflammatory cell infiltrate of the olfactory epithelium reported here.

Finally, as shown in this study the animals prefer spending more time away from the areas with the highest concentrations of $NH_3$ and that these areas overlap with the latrine area with limited spill over to adjacent areas. This is irrespective of the location of the latrine.

## Exposure limits to $NH_3$ specific for mice are missing

From our results and those of others as reported above can be concluded that a consensus on an evidence based exposure limit for mice is missing but much warranted. To date most small rodent facilities use the work place $NH_3$ exposure limit for humans (see papers and guides cited above), being an average exposure of ≤25 ppm $NH_3$ with peaks of ≤50 ppm during eight hours per day. These thresholds were derived from field and laboratory studies with human subjects, using self-reported discomfort and objective assessment of pulmonary ventilation (static/dynamic spirometry) (for a detailed recent review see EPAs Toxicological Review of Ammonia Noncancer Inhalation [CASRN 7664-41-7] Supplemental Information, September 2016). The controlled exposures studies on small rodents (rats and mice) cited above indicate that ≤56 ppm (= 40 mg/m$^3$) $NH_3$ does not inflict signs of discomfort, clinical symptoms or histopathological changes to the airways even at prolonged exposure. In contrast to humans, $NH_3$ appears not to be aversive or impact physical activity of mice even at constant exposure

levels of 100 ppm [33, 80]. Clinical symptoms as eye-blinking, and histopathological changes to the anterior compartment of the nose have been reported for controlled prolonged exposures in the range 100–150 ppm [40, 92]. Humans respond to high levels of $NH_3$ (500 ppm) with an increased pulmonary ventilation [95]. In contrast, rats and mice decrease their breathing rate (a 50% reduction at 300 ppm $NH_3$) seemingly protecting their airways from exposure to $NH_3$ [36, 37]. In summary, the existing literature of controlled exposure studies on small rodents seems to suggest an average $NH_3$ exposure of $\leq 56$ ppm with peak values $<75$–100 ppm a rational exposure limit for mice based on current knowledge.

## Concluding remarks

The objective of this study was to deepen our insights into the longitudinal changes of the cage microenvironment and the spontaneous in-cage activity of a widely used mouse strain housed in IVCs with a cage change cycle of two weeks. By gathering longitudinal data, it was possible to compare the first and second week of the cage-change cycle. By analysing repeated biweekly cage change cycles, variations across cycles could be assessed. Finally, the study was conducted in parallel at four sites in different countries within EU, enabling distinction of features which reproduced across sites from those that did not.

Cage change induced a marked increase in activity (~40%) being more pronounced during day time when the animals normally rest than during night time. The subsequent decline from this activity burst occured mainly during the first week. Our data strongly support the notion that from the animal's perspective, bi-weekly cage change is to be preferred over weekly cage change. Irrespective of the cage change frequency, the impact of a cage change is such that it must be incorporated into the experimental design as a variable.

Following a cage change, the mice quickly assigned a location(s) for the latrine(s). Females but not males showed a clear preference for having the latrine(s) in the front of the cage. In cages with males, the mice more or less instantly preferred to be active in the latrine free part of the cage floor. A behaviour that progresses through the cage change cycle. This behaviour was equally detected on both AC and CC. Reducing housing density to two mice per cage abolished it. Female mice displayed a different pattern of in-cage activity, using the entire cage floor the first week while during the second week the amount of activity in the latrine area decreased by about 5–10%, a trend that was significant only at two out of the four sites.

Measurement of $NH_3$ ppm across the cage floor revealed a gradient with three times higher values for the latrine area than the opposite area. As expected, in-cage bacterial load covaried with in-cage $NH_3$ ppm. Across the cage change cycle, $NH_3$ ppm increased from 0–1 ppm to reach ~25 ppm in the latrine free area and 50–75 ppm in the latrine area at the end of the cycle. While the latter values exceed the exposure limits set for humans, the literature on the impact of controlled $NH_3$ exposures in small rodents suggest that these limits may be too restrictive for mice.

Post hoc histopathological analysis of the nose cavity revealed mild to moderate signs of abnormalities that did not covary with the recorded in-cage $NH_3$ ppm. Seven out of the nine stigmata were also present in the germ free reference group with no lifetime $NH_3$ exposure suggesting that these may be instigated by other in-cage components than $NH_3$ such as dust or chemicals from the bedding material. Further studies on bedding materials are needed. However, an optimal solution would be a nontoxic and dust free material with properties that considerably reduce the production of $NH_3$ while meeting the demands of the mice.

We conclude that housing of four (or equivalent biomass) C57BL/6J mice for 10 weeks under the described conditions does not cause any overt discomfort to the animals.

## Supporting information

**S1 File.**
(PDF)

## Acknowledgments

We thank Dr. Inken Beck and the Animal Research Center, Ulm University, Oberberghof, Ulm, Germany, for providing the germfree mice. Gianpaolo Milite (Scientific Consultant, Udine, Italy) for organising the germfree study and critically reviewing the manuscript, Martijn Bolderheij who performed the experiments at the LUMC, the animal facility staff and animal technicians at the four sites involved in the studies, and Tapvei for providing the bedding at all four sites.

## Author Contributions

**Conceptualization:** B. Ulfhake, H. Lerat, J. Honetschlager, G. Rosati, J.-B. Prins.

**Data curation:** B. Ulfhake, K. Pernold, M. Rynekrová, K. Escot, C. Recordati, R. V. Kuiper, J.-B. Prins.

**Formal analysis:** B. Ulfhake, C. Recordati, R. V. Kuiper, G. Rosati, M. Rigamonti, S. Zordan.

**Investigation:** J.-B. Prins.

**Methodology:** B. Ulfhake, C. Recordati, G. Rosati, J.-B. Prins.

**Project administration:** K. Pernold, M. Rynekrová, K. Escot, G. Rosati, J.-B. Prins.

**Resources:** J.-B. Prins.

**Supervision:** B. Ulfhake, H. Lerat, J. Honetschlager, J.-B. Prins.

**Validation:** M. Rigamonti, S. Zordan.

**Visualization:** B. Ulfhake, M. Rigamonti, S. Zordan.

**Writing – original draft:** B. Ulfhake, H. Lerat, J.-B. Prins.

**Writing – review & editing:** B. Ulfhake, H. Lerat, J. Honetschlager, C. Recordati, R. V. Kuiper, G. Rosati, M. Rigamonti, S. Zordan, J.-B. Prins.

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
