## [Decision Letter · Decision Letter 0]

16 Nov 2021

PONE-D-21-29124A multicentre study on spontaneous in-cage activity and micro-environmental conditions of IVC housed C57BL/6J mice during consecutive cycles of bi-weekly cage-changePLOS ONE

Dear Dr. Ulfhake,

Thank you for submitting your manuscript to PLOS ONE. After careful consideration, we feel that it has merit but does not fully meet PLOS ONE’s publication criteria as it currently stands. Therefore, we invite you to submit a revised version of the manuscript that addresses the points raised during the review process.

Both reviewers find the nature of the study important and the results potentially interesting.  Although both reviewers believed the work was technically performed well, Reviewers #1 and #2 had several important comments and criticisms that need to be address.  Reviewer #1 had some minor comments, and a number of questions which should be easy for you to address in the manuscript resubmission or in comments to this reviewer.  

Reviewer # 2 had a few major criticisms.  The first regards the proper use of control groups the proper conclusions that can be drawn for this study.  Another comment of this reviewer deals reporting effect sizes and confidence estimates, I would like you to address both of these concerns,  which I believe is necessary for this work.   

Another major criticism of this manuscript by reviewer #2 deals with the interpretation of the data as presented in the discussion, and I feel in the introduction.   This review stated that you presented and weighted heavily an "optimistic interpretation" of the data.  This struck me as a serious concern, especially considering the competing  financial interests that "could reasonably be perceived as interfering with, the full and objective presentation" of the work, as three authors are employees of Tecniplast and this company funded part of the work. In response to this last criticism, I think it should be possible to present a wider range of interpretations - to achieve a more balanced discussion and avoid the perception of bias.  

I do apologize for the amount of time, taken on this review.  There have been many discussions on the nature of the perceived financial conflict and discussion of the data.  

We look forward to receiving your revised manuscript.

Kind regards,

Gregg Roman, PhD

Academic Editor

PLOS ONE

“The work at IMG was funded by IMG. The work at UGA was funded by UGA. The work at KI was funded by Karolinska Institutet and the Swedisch National Research Council (Grant 2020-02009-3). The work at LUMC was funded by the LUMC. DVC®equipment at LUMC and UGA was made available by Tecniplast SpA. S. Zordan, M. Rigamonti and G. Rosati are employed by Tecniplast SpA. Tecniplast SpA provided support in the form of salaries for authors SZ, MR and GR. SZ, MR and GR contributed to the study design, data collection and analysis, decision to publish, or preparation of the manuscript. The specific roles of these authors are articulated in the ‘author contributions’ section.”

We note that you have provided funding information within the Funding Section. Please note that funding information should not appear in other areas of your manuscript. We will only publish funding information present in the Funding Statement section of the online submission form.

 “The work at IMG was funded by IMG. The work at UGA was funded by UGA. The work at KI was funded by Karolinska Institutet and the Swedisch National Research Council (Grant 2020-02009-3). The work at LUMC was funded by the LUMC. DVC®equipment at LUMC and UGA was made available by Tecniplast SpA. S. Zordan, M. Rigamonti and G. Rosati are employed by Tecniplast SpA. Tecniplast SpA provided support in the form of salaries for authors SZ, MR and GR. SZ, MR and GR contributed to the study design, data collection and analysis, decision to publish, or preparation of the manuscript. The specific roles of these authors are articulated in the ‘author contributions’ section.”

“The authors declare no conflict of interest. Tecniplast SpA (Via I Maggio 6, 21020 Buguggiate (Va), Italy) is a commercial company selling the DVC® system. However, this does not alter the authors’ adherence to all the PLOS ONE policies on sharing data and materials. We have read the journal’s policy and the authors of this manuscript have no competing interests. The data recorded at each site are the propriety of the sites.”

Reviewers' comments:

Reviewer's Responses to Questions

**Comments to the Author**

1. Is the manuscript technically sound, and do the data support the conclusions?

Reviewer #1: Yes

Reviewer #2: Partly

2. Has the statistical analysis been performed appropriately and rigorously? 

Reviewer #1: I Don't Know

Reviewer #2: I Don't Know

3. Have the authors made all data underlying the findings in their manuscript fully available?

Reviewer #1: Yes

Reviewer #2: No

4. Is the manuscript presented in an intelligible fashion and written in standard English?

Reviewer #1: Yes

Reviewer #2: Yes

5. Review Comments to the Author

Reviewer #1: This is a very thorough paper evaluating a topic that is important from both animal welfare and scientific perspectives, and it will be a valuable addition to the literature. The approach of replicating the protocol at four different centres is especially helpful. The discussion would benefit from further consideration of the scientific, animal welfare and practical implications of your findings.

Specifically, do you feel able to make recommendations regarding potential effects on data quality if animals are used in procedures during the period of increased activity immediately following cage change? How might this be addressed via experimental design? It looks as though bi-weekly cage change is to be preferred from an animal welfare perspective; would you recommend this? If animals' activities are shifted away from the latrine, then the usable area of the cage is reduced, which will impact on welfare. Does this have implications for minimum cage dimensions for IVCs (and/or conventional caging)? Can you comment on whether human exposure limits are acceptable for mice; is there evidence for this or is it an assumption that may or may not be appropriate for other animals? These are the kinds of questions and issues that ran through my mind when I read the manuscript, and it would make the paper more impactful to draw these out.

A couple of minor points: could you use the term 'litter' instead of 'bedding'? Rodents cannot make a 'bed' out of litter, nor can they 'burrow' in it (line 170; 'digging' would be more accurate).

Some of the English and spelling needs a bit of attention, e.g. 'an eye need to be kept' (line 99) should be 'needs' but actually it would be better to say those parameters need to be monitored. Line 106 'animal's welfare and health' should be 'the welfare and health of the animals'; 'scaring' instead of 'scarring' in lines 617 and 826 etc.

Reviewer #2: The authors addressed the effects of a bi-weekly cage change interval in IVC-housed mice on home cage behaviour, ammonia concentrations and upper airway histopathology. These are important issues, especially since IVCs are typically adopted to extend cage-change intervals, although current evidence indicates that they are not able to limit ammonia levels sufficiently to avoid nasal lesions in both breeding and stock cages. Moreover, mice find the high ventilation rates of IVCs aversive, which is associated with heightened levels of fear, anxiety, and stress compared to mice in conventional caging.

Major comments

I have two major concerns about this study. First, although the authors aimed at studying the effect of a bi-weekly cage change procedure, they did not include any control treatments. Thus, neither shorter nor longer cage change intervals were studied to compare the effects with. Instead, to asses the effects on upper airway histopathology, they compared their study mice with mice from a completely independent population of mice, which they called "baseline control". However, this is not a proper control group, and direct comparison of measures is inappropriate. Thus, the authors may describe behaviour, ammonia levels, and airway histopathology in their study animals, and compare them between centres, but they are unable to draw formal inferences regarding the effects of shorter or longer cage change intervals, or non-IVC housing conditions.

Secondly, the interpretation of the results are overly optimistic. Thus, they conclude that up to 4 females and up to 3 males can be housed under these conditions without exceeding the ammonia exposure limits for humans. However, their data show that with time since the last cage change, the mice increasingly avoid the area of their latrine (half of the cage size!) where, towards the end of the 2-week period, ammonia levels are measured that may be seriously detrimental to health. Thus, by letting feaces and urine accumulate for so long, the usable space for the mice is increasingly restricted (which is mirrored by the gradually decreasing activity), thereby effectively violating the minimal space requirements for laboratory mice. This is a less optimistic interpretation of the same results, and the authors should carefully adress these considerations when revising the current manuscript.

The authors excessively use infernetial statistics (p-values) although this is an exporatory study that has not been designed for hypothesis tests. Thus, they should present their results in terms of effect sizes and confidence estimates, rather than hypothesis tests.

Minor comments:

The title is very confusing, unclear what exactly the study was about; the term multicentre study may actually be misleading as there was no treatment that was tested across multiple centres but centre was the main independent variable. Perhaps better write: "Between-laboratory variation in activity and ammonia concentrations in IVC-housed C57BL/6J mice with bi-weekly cage changes". "spontaneous in-cage" is unnecessary and "micro-environmental conditions" is unclear and somewhat exaggerated given that only ammonia was measured.

L.80 behaviour without –s; it is not the behaviours (behavioural elements) that are altered but the behaviour (i.e. the organisation of behavioural patterns in space and time). And what about aggression? One of the most obvious changes in behaviour after a cage change is enhanced aggression in male mice.

L.96 use either ammonia OR NH3 throughout

L.135-136 how can IMG serve as a control site for comparisons of in-cage activity? If four centres measure in-cage activity, how can one site be a control site?

L.161 can you specify the method of randomization?

L.169-170 what do you consider as burrowing activities in loose bedding material?

L.180-181 the authors are concerned about anxiety induced by the cage change but don't seem to consider tunnel-handling or cup-handling instead of tail handling, which is known to induce stress and anxiety compared to less aversive handling methods. Similarly, the light-dark cycle is such that animals are disturbed by husbandry procedures during their main resting periods.

L.278 What is a baseline control? These mice are from a completely different population of mice and, therefore, are not a proper control to assess the effects of the bi-weekly cage change procedure. It is unclear how they were raised, housed, and cared for, and no mention is made about the ammonia levels they may have been exposed to. This needs to be clarified.

L.688-690 this statement is inappropriate as your study did not include a treatment with weekly cage changes.

L.736 the authors consider the shift in activity away from the soiled area as evidence "demonstrating the effectiveness of the IVC working principle". A less optimistic interpretation suggests that IVCs don't work as they are unable to prevent the accumulation of ammonia and hence increasingly constrain the usable space of the mice.

6. PLOS authors have the option to publish the peer review history of their article (what does this mean?). If published, this will include your full peer review and any attached files.

Reviewer #1: No

Reviewer #2: No

---

## [Author Response · Author response to Decision Letter 0]

1 Mar 2022

Based on the issues raised by the reviewers, the major revisions include Abstract, section added to the Introduction expanding the explanation of the rational for the study and its design including use of the axenic animals. Expanded statistical section in Material and methods. Addition of effect size measures where appropriate and these were missing in the original submission. In the discussion, a section about use of floor space (issue raised by both reviewers) and the relevance of the NH3 ppm exposure limits set for humans (issue raised by reviewer #1) have been added. In addition, the concluding remarks have been rewritten in the spirit of helping the reader to extract the key outcomes of the study. Furthermore, a rather large number of minor edits have been conducted throughout the text (indicated as tracked changes).

The resubmission package contains along with this cover letter:

Revised manuscript with track-changes as a .doc

Revised manuscript w.o. track-changes as a .doc

Revised manuscript w.o. track-changes compiled with Tables and Figs as a PDF

Updated set of Figures (TIFF format)

Tables as PDFs

Updated and complemented Supportive information as PDF

Major points:

1. As stated by the authors, all except three of the authors have independent academic positions with no commercial links to Tecniplast. The Tecniplast company provided support in the form of salaries to three authors but did not influence or partake in the study design, data collection and analysis, decision to publish, or preparation of the manuscript. The results and records remain the intellectual property of each participating academic site. All sites can rightfully be considered as major sites within EU for housing and use of laboratory animals in the Life Sciences.

2. The objective and design of this study was not to compare bi-weekly cage change with weekly cage change. According to the literature, the cage-change period may vary between twice a week to once every 28 days or to be “on demand” i.e., without a fixed interval. The aspiration of this study was to deepen our insights into the longitudinal changes of the cage microenvironment and the spontaneous in-cage activity of a widely used mouse strain housed in IVCs with a cage change cycle of two weeks. By gathering longitudinal data, it is possible to compare the first and second week of the cage-change cycle. By analysing repeated biweekly cage change cycles, variations across cycles can be assessed. The study was conducted in parallel at four sites in different countries within EU, enabling distinction of features which reproduced across sites from those that did not. 

The provided data of in-cage rest and physical activity is a unique record of every day in-cage life and how procedures affect it. As such it may prove valuable along with other metrics in future discussions on decisions about regulations of minimal cage floor area for small rodents.

3. Reviewer #2 commented on the use of GF animals as controls. It was not our intention to present this cohort as controls but as a reference group. We have rephrased this in the revised manuscript. Available data (see cited literature in the revised manuscript) indicate that GF animals generate less NH3 in their intestinal tract and have lower serum levels of NH3 than wild type mice because of the absence of a gut microbiota. The lack of microbiota in their fecal deposits will also hamper any production of NH3 from urea through bacterial ureases. In our opinion, age, sex, and strain matched GF animals with nil lifetime exposure to NH3 are therefore appropriate reference for the cohorts in our study that have been exposed to varying concentrations of NH3 throughout their lifetimes. Further details on the origin and maintenance of the axenic animals have been provided as supportive information.

4. Effect size has been added along with the p values (see Figs 6a-b, 7a-b, 8a-b, 11a-b, 12a-b, 14a-b, 15a-b, 18a). Effect size measures are straightforward with data sets that are homoscedastic (having equal variance) and follows a normal distribution. In complex parametric models, mixed models and non-parametric statistics, effect size measures are less intuitive. Data sets analysed in our study did not follow a normal distribution, accordingly we used nonparametric statistics and have added the recommended effect size measures for the different tests (described with relevant citations under Statistics in Material and methods) to the revised manuscript. Nonparametric effect size measures in contrast to parametric effect size measures consider only if A is equal, larger, or smaller than B, not how large the difference between A and B is. In the rank correlations and the mixedPCA presented in the original manuscript, rs and vector length and direction are the effect size measures.

5. We have carefully reassessed our claims and conclusions in view of our experimental results as suggested by the second reviewer. Phrases that may have been perceived as biased have been deleted or rephrased.

Additional minor comments:

1. We believe that we have added to the Discussion some answers to the questions asked by reviewer #1. We have corrected indicated spelling errors.

2. DVC can only inform about rest and physical activity (and level of physical activity). Furthermore, data is only collected from the cage floor. Thus, we are not able to comment on any specific behaviour or activity above the floor.

3. L 80 we have made the change. 

4. L 96 – Throughout the manuscript, we have replaced ammonia by NH3

5. L 161 – methods of randomisation have been specified in the text and supplemental information. 

6. L 169-170 – burrowing has been replaced by digging.

7. L 180-181 – We agree with the reviewer that tunnel-handling and cupping are preferred over tail handling. We used tail handling at all sites hence is not a variable in our study and is therefore expected to contribute equally to the anxiety at all centres. A comparison between handling method and the effects of cage change could be a subject of a follow-up study. Husbandry procedures were limited to the daily checks and the cage changes. 

8. L 278 – see under major comments

9. L 688-690 – statement has been changed. 

10. L 736 – text has been changed.

---

## [Decision Letter · Decision Letter 1]

6 Apr 2022

A multicentre study on spontaneous in-cage activity and micro-environmental conditions of IVC housed C57BL/6J mice during consecutive cycles of bi-weekly cage-change

PONE-D-21-29124R1

Dear Dr. Ulfhake,

We’re pleased to inform you that your manuscript has been judged scientifically suitable for publication and will be formally accepted for publication once it meets all outstanding technical requirements.

Kind regards,

Gregg Roman, PhD

Academic Editor

PLOS ONE

Additional Editor Comments (optional):

Reviewers' comments:

Reviewer's Responses to Questions

**Comments to the Author**

1. If the authors have adequately addressed your comments raised in a previous round of review and you feel that this manuscript is now acceptable for publication, you may indicate that here to bypass the “Comments to the Author” section, enter your conflict of interest statement in the “Confidential to Editor” section, and submit your "Accept" recommendation.

Reviewer #1: (No Response)

2. Is the manuscript technically sound, and do the data support the conclusions?

Reviewer #1: Yes

3. Has the statistical analysis been performed appropriately and rigorously? 

Reviewer #1: I Don't Know

4. Have the authors made all data underlying the findings in their manuscript fully available?

Reviewer #1: Yes

5. Is the manuscript presented in an intelligible fashion and written in standard English?

Reviewer #1: Yes

6. Review Comments to the Author

Reviewer #1: I can see that you have addressed the comments that were easier to deal with, but I'm disappointed that you still didn't take the opportunity to identify and discuss more of the animal welfare issues.

7. PLOS authors have the option to publish the peer review history of their article (what does this mean?). If published, this will include your full peer review and any attached files.

Reviewer #1: No

---

## [Editor Report · Acceptance letter]

10 May 2022

PONE-D-21-29124R1 

A multicentre study on spontaneous in-cage activity and micro-environmental conditions of IVC housed C57BL/6J mice during consecutive cycles of bi-weekly cage-change 

Dear Dr. Ulfhake:

I'm pleased to inform you that your manuscript has been deemed suitable for publication in PLOS ONE. Congratulations! Your manuscript is now with our production department. 

Kind regards, 

on behalf of

Dr Gregg Roman 

Academic Editor

PLOS ONE